

**Conservation agriculture increases soil organic carbon stocks but not soil CO₂ efflux in**
**two 8-year-old experiments in Zimbabwe**
Armwell Shumba[a,b,c*], Regis Chikowo[a,d], Christian Thierfelder[e], Marc Corbeels[f,g], Johan Six[h],
Rémi Cardinael[a,b,f]
[a]Department of Plant Production Sciences and Technologies, University of Zimbabwe, Harare,
Zimbabwe
[b]CIRAD, UPR AIDA, Harare, Zimbabwe
[c]Fertilizer, Farm Feeds and Remedies Institute, Department of Research and Specialist
Services, Ministry of Lands, Agriculture, Fisheries, Water and Rural Development, Harare,
Zimbabwe
[d]Plant, Soil and Microbial Sciences Department, Michigan State University, East Lansing, MI
48824, USA
[e]International Maize and Wheat Improvement Center (CIMMYT), P.O. Box MP 163, Mount
Pleasant, Harare, Zimbabwe
[f]AIDA, Univ Montpellier, CIRAD, Montpellier, France
[g]IITA, International Institute of Tropical Agriculture, PO Box 30772, Nairobi, 00100, Kenya
[h]Department of Environmental Systems Science, ETH Zurich, 8092 Zürich, Switzerland
* Corresponding author. Email: armwellshumba123@gmail.com



**Abstract**

Conservation agriculture (CA), combining reduced or no tillage, permanent soil cover and
improved rotations, is often promoted as a climate-smart practice. However, our understanding
about the impact of CA and its respective three principles on top and sub-soil organic carbon
(SOC) stocks and on soil $CO_2$ efflux in low input cropping systems of sub-Saharan Africa is
rather limited. The study was conducted at two long-term experimental sites established in
2013 in Zimbabwe. The soil types were abruptic Lixisols at Domboshava Training Centre
(DTC) and xanthic Ferralsol at the University of Zimbabwe farm (UZF). Six treatments,
replicated four times were investigated: conventional tillage (CT), conventional tillage with
rotation (CTR), NT, no-tillage with mulch (NTM), no-tillage with rotation (NTR), no-tillage
with mulch and rotation (NTMR). Maize (*Zea mays* L.) was the main crop and treatments with
rotation included cowpea (*Vigna unguiculata* L. Walp.). SOC concentration and bulk density
were determined for samples taken from the 0-5, 5-10, 10-15, 15-20, 20-30, 30-40, 40-50, 50-
75 and 75-100 cm depths. Gas samples were regularly collected using the static chamber
method during the 2019/20 and 2020/21 cropping seasons and during the 2020/21 dry season.
SOC stocks were significantly ($p < 0.05$) higher under NTM, NTR and NTMR compared to
NT and CT in top 5 and 10 cm layers at UZF, while SOC stocks were only significantly higher
under NTM and NTMR compared to NT and CT in top 5 cm at DTC. NT alone had a slightly
negative impact on top SOC stock. Cumulative SOC stocks were not significantly different
between treatments when considering the whole 100 cm soil profile.  Regardless of larger
organic carbon inputs in mulch treatments, there were no significant differences in $CO_2$ efflux
between treatments, but it was higher in maize rows than in inter-rows as a result of autotrophic
respiration from maize roots. Our results show the overarching role of crop residue mulching
in CA cropping systems in enhancing SOC storage but that this effect is limited to the topsoil.



**Key words:** climate change mitigation, climate-smart agriculture, deep soil organic carbon,
mulch, sustainable intensification

**1. Introduction**
Soil organic carbon (SOC) is an important determinant of soil fertility, productivity and
sustainability, and is a useful indicator of soil quality in tropical agricultural systems where
nutrient poor and highly weathered soils are managed with little external inputs (Chivenge et
al., 2007; Feller & Beare, 1997; Lal, 1997). Therefore, rebuilding depleted SOC stocks in such
soils holds potential to contribute to climate change mitigation (Bossio et al., 2020; Minasny
et al., 2017; Swanepoel et al., 2016) through sustainable management of agricultural soils
(Dignac et al., 2017; Paustian et al., 2016).
Conservation agriculture (CA), based on minimum soil disturbance, crop residue retention and
crop rotation, has been known to improve surface SOC, with beneficial effects on soil
functioning such as improved water infiltration (Thierfelder & Wall, 2009, 2012) and better
aggregate stability (Six et al., 1999; Thierfelder & Wall, 2012). The potential of CA to increase
SOC stocks and thereby mitigate climate change has, however, been much debated (Corbeels,
Cardinael, et al., 2020). The general understanding is that, this potential is relatively low (Du
et al., 2017; Powlson et al., 2014), which is well demonstrated in sub–Saharan Africa (SSA)
(Cheesman et al., 2016; Corbeels et al., 2019; Powlson et al., 2016). In fact, soil C storage has
often been over-estimated for CA due to shallow soil sampling. Compared to conventional
tillage systems, no-tillage redistributes SOC in the soil profile, with higher concentrations in
the topsoil but potentially lower concentrations below, which can result in no differences in
whole profile SOC stocks between no-tillage and conventional tillage (Angers & Eriksen-
Hamel, 2008). However, this lack of significant differences in many studies assessing whole

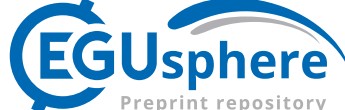

profile SOC stocks suffer from not enough statistical power to accurately assess the potential
significant SOC changes (Kravchenko & Robertson, 2011).
CA can potentially build SOC in deeper soil layers from e.g. the use of cover crops with more
extended root systems (Kell, 2011; Thorup-Kristensen et al., 2020; Yang et al., 2023).
However, proper soil sampling strategies, to account for both topsoil and subsoil (> 30 cm
depth) SOC stocks must, therefore, be prioritized. Soil sampling has often been limited to the
top soil plough layer (0-30 cm) in the past two decades (Dube et al., 2012; Patra et al., 2019;
Powlson et al., 2016; Yost & Hartemink, 2020), where SOC concentrations, root densities
(Chikowo et al., 2003) and microbial activities (Mtambanengwe et al., 2004) are generally
largest  (Rumpel et al., 2012) and which is the minimum default soil depth recommended by
the Intergovernmental Panel for Climate Change (IPCC, 2019). In a meta-analysis of SOC
stocks in the top 1 m of soils, Balesdent et al. (2018) found that soils below 0.3 m contain on
average 47 % of total SOC stock in the 1 m soil depth and accounts for 19 % of the SOC that
has been recently incorporated. Therefore, focusing on topsoil only, could underestimate the
potential of agricultural management practices to store SOC (Cardinael et al., 2015). In turn,
this can give wrong conclusions on the climate change mitigation potential of agricultural
management practices.
There has been many studies on the effects of CA on crop productivity and soil health benefits
(Corbeels, Naudin, et al., 2020; Kimaro et al., 2016; Mhlanga et al., 2022a; Swanepoel et al.,
2018; Thierfelder et al., 2015, 2017; Thierfelder & Mhlanga, 2022), and other studies have
fuelled the debate on CA practicality and adoption in SSA (Giller et al., 2009, 2015; Kassam
et al., 2019). However, the effects of CA on SOC dynamics and soil $CO_2$ efflux have not been
widely investigated in SSA. Thierfelder et al., (2017) have alluded to the fact that, data on
climate change mitigation potential of CA in southern Africa is scanty hence the need for more



research to better quantify the mitigation effects of CA as a climate-smart technology. It has
also been observed that depending on the socio-economic and biophysical conditions, farmers
may find it easier to adopt certain CA principles and/or their different combinations (Baudron
et al., 2012; Mbanyele et al., 2021), although this also opened up new debates (Thierfelder et
al., 2018). Therefore, in this study, the focus was on the individual versus combined effects of
CA principles (no-tillage, crop residue retention, crop rotation) on SOC stocks and soil $CO_2$
effluxes.
As changes in SOC stocks take time to be detected, long-term experiments are crucial, but are
rare, especially in Africa (Bationo et al., 2013; Cardinael et al., 2022; Powlson et al., 2016;
Thierfelder & Mhlanga, 2022). This study was conducted on two long-term experiments
established in 2013 in Zimbabwe. We hypothesized that the full combination of CA
components would be associated with more rapid increases of SOC stocks and soil $CO_2$ efflux
than adoption of only one component, and that this increase would mainly be due to increased
C inputs to the soil and minimum soil disturbance.

## 2. Materials and methods

### 2.1 Study sites

The study was conducted at two long-term experimental sites established in November 2013
by CIMMYT. The site at the University of Zimbabwe Farm (UZF) is located about 12 km north
of Harare city centre (31° 00′ 48″ E; 17° 42′ 24″ S), while the site at the Domboshava Training
Centre (DTC) is located about 30 km north-east of Harare (31° 07′ 33″ E; 17° 35′ 17″ S). UZF
soils are dolerite-derived xanthic *Ferralsols* (FAO classification) and are medium-textured
sandy clay loams (34 % clay) in the top 20 cm with a subsoil (20-40 cm) of slightly higher clay



content (38 %). DTC soils are granite-derived abruptic *Lixisols* (FAO classification) and are
light-textured sandy loams (15 % clay) in the 0-20 cm layer, overlying abruptly a heavier-
textured subsoil (20-40 cm) of 30 % clay (Figure 1).

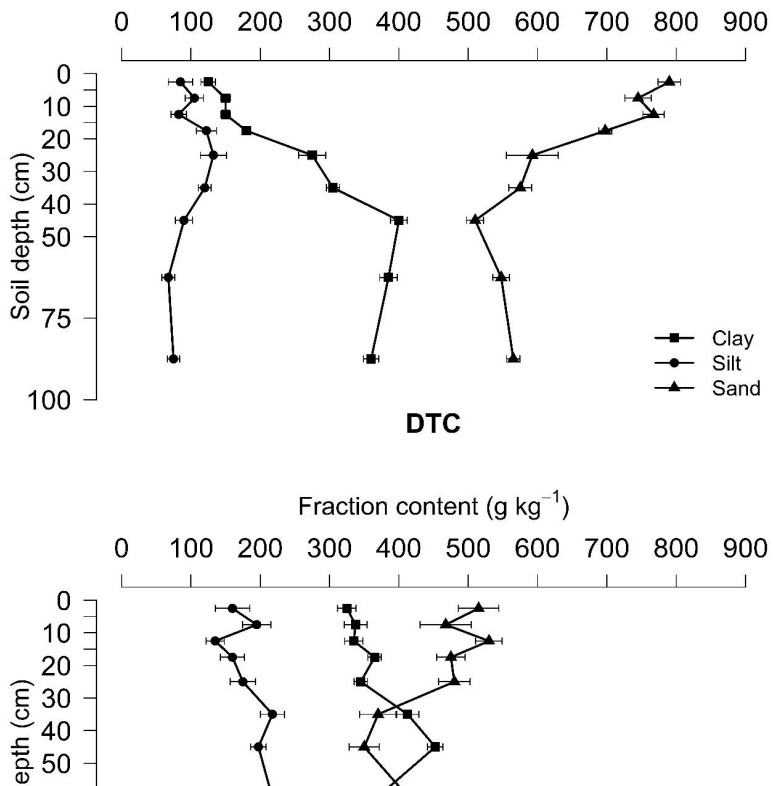

**Figure 1**: Soil texture to 1 m soil depth at the DTC (top) and UZF (bottom) sites in Zimbabwe.
Error bars represent standard errors (N = 4).



The two study sites have a sub-tropical climate with cool, dry winters and hot, wet summers
with mean annual minimum and maximum temperatures of 12°C and 25°C, respectively
(Mapanda et al., 2010). The rainy season starts in November and tails off in March with a mean
annual rainfall of 826 and 814 mm at UZF and DTC, respectively (Mhlanga, et al., 2022b).
Cumulative seasonal rainfall in 2019/20 (474 mm) was almost half of rainfall received in the
2020/21 season (932 mm) at DTC (Figure S1). At UZF, cumulative seasonal rainfall was 551
mm and 637 mm in the 2019/20 and 2020/21 cropping seasons. On average, minimum and
maximum temperatures were 16.9 and 28.1 °C in 2019/20 and 15.5 and 27.2 °C in 2020/21 at
DTC and 15.5 and 28.6 °C in 2019/20 and 15.3 and 27.5 °C in 2020/21 cropping season at
UZF.

**2.2 Experimental treatments and crop management**
Two identical experiments were set up at the study sites and treatments were maintained every
season since November 2013. The experiments were set up in a randomised complete block
design (RCBD) with eight treatments replicated in four blocks. However, in this study we
investigated only six of these treatments. All crop residues were removed soon after harvesting
in all treatments, stored and then applied prior to planting in treatments with mulch. The six
treatments in our study were:

i.   Conventional tillage (CT) – land preparation was done through digging with a hand hoe

and maize (*Zea mays* L.) was sown as a sole crop in rip lines that were created

afterwards using an animal-drawn Magoye ripper (a traditional plough with the

mouldboard replaced with a ripper tine) at DTC, and in planting basins (approximately

10 cm diameter and 10 cm depth) created using a hand hoe at UZF.



ii.    Conventional tillage with rotation (CTR) – land preparation was done as in the CT

treatment and maize was rotated with cowpea (*Vigna unguiculata* L.).

iii.    No-tillage (NT) – sole maize was sown in rip lines created using an animal-drawn

Magoye ripper (no further soil disturbance was done) at DTC, and in planting basins

(approximately 15 cm diameter and 15 cm depth) created using a hand hoe at UZF.

iv.    No-tillage with mulch (NTM) – maize was sown as in the NT treatment and maize

residues from the previous season were applied on the soil surface between maize

rows at planting at a rate of 2.5 t DM $ha^{-1}$.

v.    No-tillage with rotation (NTR) – maize was sown in rip lines and rotated with cowpea.
vi.    No-tillage with mulch and rotation (NTMR) – maize was sown in rip lines and rotated

with cowpea and maize residues were applied on the soil surface between maize rows

at planting at a rate of 2.5 t DM $ha^{-1}$.

Crop residues were removed every year after harvest and weighed in again to maintain the
exact 2.5 t $ha^{-1}$ residue weight year after year. There was a total of 24 plots at each site which
were 6 m wide and 12 m long (72 $m^2$). Treatments with rotation (CTR, NTR, NTMR) were
split into 6 m wide and 6 m long (36 $m^2$) subplots where maize and cowpea were grown
interchangeably every season (maize was sown on one side of the plot while cowpea on the
other side).
The inter-row spacing was 90 cm and 45 cm for maize and cowpea, respectively, and the in-
row spacing was 25 cm for both crops which translated to 44,444 and 88,888 plants $ha^{-1}$,
respectively. Three seeds were planted per planting station and thinned to one after emergence.
Basal dressing of nitrogen (N), phosphorus (P) and potassium (K) was applied within 5 cm of
the seeds in the form of compound fertilizer for both maize and cowpea at 11.6 kg N $ha^{-1}$, 10.6
kg P $ha^{-1}$ and 9.6 kg K $ha^{-1}$, respectively. Nitrogen top dressing to maize only, was applied at



4 and 8 weeks after emergence (WAE) in two equal splits of 23.1 kg N ha⁻¹ each, as ammonium
nitrate when soil moisture was adequate. However, in the 2019/20 cropping season at both sites
and in the 2020/21 cropping season at UZF, the first N top dressing was delayed by an average
of 4.5 and 2.0 weeks, respectively, due to mid-season dry spells. Ammonium nitrate was side
dressed within 5 cm of the maize stems. Glyphosate [N-(phosphono-methyl) glycine], a pre-
emergent non-selective herbicide was applied at 1.025 L active ingredient ha$^{-1}$ soon after
sowing to control weeds. This was followed by manual hoe weeding whenever weeds reached
a maximum of 10 cm height or 10 cm in diameter for stoloniferous weeds to achieve a weed
clean field. More details about the experiment can be found in Shumba et al., (2022) and
Mhlanga et al., (2022a).

**186  2.3 Soil sampling for bulk density determination and soil organic carbon analysis**

Soil sampling was done in May 2021 at both sites. For each treatment and replicate, two
sampling points in the maize rows and two sampling points in the middle of the inter-rows
were randomly selected. The two samples from the rows were pooled into one sample per
depth, similarly to the two samples taken in the inter-rows. The following nine depth
increments were considered for both SOC and bulk density (BD) measurements: 0-5, 5-10, 10-
15, 15-20, 20-30, 30-40, 40-50, 50-75 and 75-100 cm. Undisturbed soil samples using a metal
cylinder (5 cm diameter and 5 cm height) were taken from the following depth ranges 0-5, 5-
10, 10-15, 15-20 and 20-30 cm for both SOC and BD measurements.  A motorized, hand-held
soil corer was used to take samples for the 30-40, 40-50, 50-75 and 75-100 cm depths for SOC
analysis from the same positions where undisturbed samples were taken. As no significant
differences in BD were found below 20 cm between the different treatments at the two sites
(see results section) and to avoid too much destruction of the experimental plots, two soil pits





were opened at the edges of the experimental plots (also cropped with maize since 2013) at
each site to take BD samples for the 30-40, 40-50, 50-75 and 75-100 cm depths. As a result,
BD below 30 cm depth was assumed the same across the treatments.
Soil samples were crumbled and fresh weight was determined using a field scale. Soil moisture
was determined on a sub-sample by drying it in an oven at $105^{\circ}$C for 48 hours. All samples
were then air-dried and sieved through a 2 mm sieve to determine the mass proportion of coarse
soil (> 2 mm) as a fraction of the whole sample. Bulk density (BD) was determined by dividing
the dry mass of soil by the volume of the cylinder. Subsamples from the ≤ 2 mm soil fraction
were grinded to < 200 µm for SOC analysis. SOC concentration was analysed with a CHN
elemental analyser.

**2.4 Soil organic carbon stocks calculation**

The equivalent soil mass (ESM) approach was used to determine SOC stocks to avoid
systematic bias in SOC calculation when using the fixed depth method (Ellert & Bettany, 1995;
von Haden et al., 2020; Wendt & Hauser, 2013). We defined reference soil mass profiles for
each site, based on the lowest cumulative soil mass obtained for each replicate. For these
references, cumulative soil mass layers were 0-650, 650-1340, 1340-2060, 2060-4160, 4160-
5590, 5590-7040, 7040-10550, 10550-13770 Mg soil ha$^{-1}$ at DTC and 0-460, 460-870, 870-
1330, 1330-1840, 1840-2760, 2760-4030, 4030-5300, 5300-8190, 8190-11050 Mg soil ha$^{-1}$ at
UZF, which roughly corresponded to soil depth layers of 0-5, 5-10, 10-15, 15-20, 20-30, 30-
40, 40-50, 50-75, 75-100 cm, respectively. SOC stocks were calculated on the basis of the same
soil mass as the reference profile but different soil depth layers which varied by < 1.5 and <
0.6 cm at DTC and UZF, respectively. As mulch was only applied between maize rows, and
fertilizer was only applied on maize rows, it was estimated that the row and interrow space

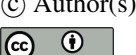



represented 22 and 78 % respectively, hence SOC calculations were weighted accordingly
(Shumba et al., 2022). Change in cumulative SOC stock between treatments for a given soil
depth was determined using the CT treatment as the reference treatment:
$\Delta SOC\ stock = SOC\ stock_{treatment(i)} - SOC\ stock_{CT(i)}$, (Equation 1)
where SOC stock$_{treatment}$ is the cumulative SOC stock per treatment (CTR, NT, NTM, NRT,
NTMR) at a given soil layer and (*i*) representing 0-5, 0-10, 0-15, 0-20, 0-30, 0-40, 0-50, 50-
75, 75-100 cm.
SOC accumulation or loss rates (kg C ha$^{-1}$ yr$^{-1}$) were calculated by dividing the change in
stocks by the number of years between the establishment of the experiment and the time of soil
sampling (8 years):
$SOC\ accumulation/loss\ rate = \frac{\Delta SOC\ stocks}{8} \times 1000$, (Equation 2)

**2.5 Estimation of organic carbon inputs to the soil**
Maize and cowpea yield and aboveground biomass were measured since the inception of the
experiment, except for cowpea during the 2013/14 season. This data gap was filled by using
the average cowpea yield and aboveground biomass values across seasons (from 2013/14 to
2020/21). We assumed that 5 % of the maize aboveground vegetative biomass remained in the
field because maize stalk slashing at harvesting did not remove the whole stem. A root:shoot
ratio of 0.16 and 0.06 for maize and cowpea, respectively (Amos & Walters, 2006; Kahn &
Schroeder, 1999; Kimiti, 2011) was used to estimate the contribution of roots to organic C
inputs to the soils. Organic C input contribution from weeds was assumed insignificant since
there was effective control of weeds through the use of pre-emergence herbicide (glyphosate)



and timely manual weeding throughout the cropping season. We also assumed that the relative
amounts of organic C transferred through rhizodeposition was the same for maize and cowpea
(i.e. 0.45 x root C biomass (Balesdent et al., 2011) and that the organic C content of all plant
parts was 430 g kg$^{-1}$ (Ma et al., 2018). Cumulative organic C inputs to the soil were then
estimated for each treatment (Cardinael et al., 2022).

**2.6 Gas sampling, analyses and flux determinations**
The static chamber methodology was used for $CO_2$ gas sampling. The static chambers had PVC
base rings (height = 0.1 m and inside radius = 0.1 m) and PVC cylindrical lids (height = 0.2 m
and inside diameter = 0.1 m). Base rings were semi-permanently driven 0.07 m into the soil to
avoid possible gas leakages and contamination by lateral diffusion (Abalos et al., 2013; Clough
et al., 2020). The lids had an airtight and self-sealing rubber septum on top through which gas
was sampled. During gas sampling, the lids were inserted about 0.02 m into the base rings and
the contact area between the base rings and the lids was always smeared with petroleum jelly to
avoid possible leakages of trapped gas. The static chambers were painted white to minimize
temperature changes in the chamber headspace from the sun's radiative heat.
Surface area coverage for each chamber was 0.0314 m$^2$ and headspace volume of 0.006 m$^3$. Gas
sampling was done simultaneously in the row and interrow spaces, each replicate having a
chamber in the row and in the middle of the inter-row (Shumba et al., 2022). It should be noted
that, $CO_2$ measured in this study consisted of effluxes coming both from autotrophic and
heterotrophic respiration.
A 20 mL syringe was used to collect gas samples at time 0 (immediately after securing the
chamber) and after 48 minutes of gas trapping. The gas samples were pressurised into pre-



evacuated 12 mL Exetainer glass vials (Labco Ltd., Lampeter SA48, United Kingdom). Linearity
tests were carried out at both sites by collecting gas samples at times 0, 15, 30, 48 and 60 minutes
of gas trapping. Results showed that $CO_2$ emissions increased linearly with time, suggesting that
two gas samplings at 0 and 48 minutes were relevant for this study since no saturation was
observed (data not shown). Gas sampling was done between 10 am and 12 pm on every sampling
day.
$CO_2$ efflux measurements were carried out during the cropping season (November to April) in
2019/20 and 2020/21, but in 2021, $CO_2$ efflux measurements were extended into the dry season
(May to September). Gas sampling was done at least every two weeks during the cropping
season, with additional sampling following fertilizer applications and rainfall events (Shumba et
al., 2022).
$CO_2$ was quantified at ETH Zurich by gas chromatography using the thermal conductivity
detector and $CO_2$ fluxes were calculated as the differences in concentration between the 0 and 48
minutes sampling times:
$$F = \frac{(GC_f - GC_o) \times V}{T \times A} \quad\quad , \text{(Equation 4)}$$

where $F$ is the gas flux (mg $CO_2$ m$^{-2}$ hr$^{-1}$), $GC_f$ and $GC_o$ are the gas concentration (ppm) at end
(time 48 minutes) and start (time 0 minutes) of chamber closure, V is the chamber volume (mL),
$T$ is the duration of the chamber closure (hours) and $A$ is the surface area covered by the static
chamber (m$^2$).

**2.7 Cumulative soil $CO_2$-C emissions**
Cumulative $CO_2$-C emissions were determined using linear interpolation between sampling
points by multiplying the mean flux of two successive sampling dates by the length of the



period between sampling and adding that amount to the previous cumulative total (Dorich et
al., 2020). Cumulative efflux per treatment was computed as the weighted contribution from
row and inter-row effluxes (Shumba et al., 2022).

**2.8 Data analysis**
Statistical analyses were performed using R software, version 4.0.0 (R Core Team 2020). Prior
to analysis, $CO_2$ data were checked for normality by both visual inspection (Quantile-Quantile
plots and density distributions) and with the Shapiro-Wilk test. Linear mixed effect models were
fitted to daily $CO_2$ emissions using the *lmer* function from the *lme4* package (Bates, 2010), using
as fixed effects the site (DTC, UZF), the season (2019/20, 2020/21), the treatment (CT, CTR,
NT, NTM, NTR, NTMR) and the chamber position (row vs inter-row). The chamber number
nested in the replicate was considered as random factor. The final models were chosen based on
the lowest Akaike information criterion (AIC) and on the lowest Bayesian information criterion
(BIC). An analysis of variance (ANOVA) was then done on the fitted models. Separation of
means was done using the post hoc Tukey test at 5 % significance level using the *emmeans*
function from the *emmeans* package (Bolker et al., 2009).
For soil data, normality was tested by the Kolmogorov-Smirnov test. After confirming that data
were normally distributed, analyses of variance (ANOVA) was carried out to establish any
significant treatment effects on BD, SOC concentration, and SOC stock. Subsequent mean
separation was done using Tukey's test.
**3. Results**
**3.1 Soil bulk density**



The different cropping systems (CT, CTR, NT, NTM, NTR, NTMR) had no significant (p >
0.05) effect on BD across all soil depths except in the 5-10 cm depth in the inter-row at DTC
(Figure 2) where BD was on average 5 % lower in the conventional tillage treatments (CT, CTR)
than in the no-tillage treatments (NT, NTR). However, soil depth and location (row or inter-row),
and the soil depth x location interaction had significant (p < 0.001) effects on BD. In the tillage
layer (0-15 cm), BD was at least 2 % higher in the inter-rows than in the rows at both sites. In
the deeper soil layer (15 – 30 cm), there were no significant (p > 0.05) differences. BD for depths
below 30 cm were the same across treatments since it was determined from pits outside the
experiment. It ranged between 1.47 – 1.51 and 1.47 – 1.49 g cm$^{-3}$ (Table S1) in the subsoil (30 –
100 cm layers) at DTC and UZF, respectively.

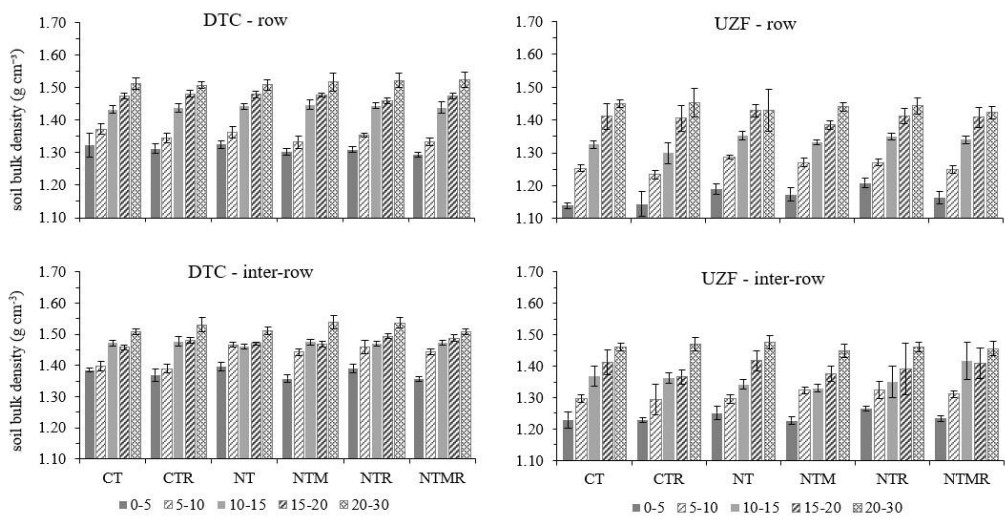


**Figure 2.** Top soil bulk density (0-30 cm) at the Domboshava Training Centre (DTC) and University of Zimbabwe Farm (UZF) experimental sites in Zimbabwe. CT: conventional tillage, CTR: conventional tillage with rotation, NT: no-tillage, NTM: no-tillage with mulch, NTR: no-tillage with rotation, NTMR: no-tillage with mulch and rotation.  Error bars represent standard errors (N = 4).



## 3.2 SOC concentrations

SOC concentration decreased significantly ($p < 0.001$) with soil depth (Figure 3, Table S2) and was highest in the surface soil (0-5 cm) for all treatments (Table S2). There were significant treatment effects in the 0-5 cm ($p = 0.001$) and 5-10 cm ($p = 0.005$) soil layers at DTC and in the 0-5 cm layer ($p < 0.001$) only, at UZF. NTM had significantly ($p < 0.05$) higher SOC concentration compared to conventional tillage treatments (CT, CTR) and NT in the 0-5 cm soil layer at both sites (Figure 3); the increase in SOC concentration ranged between 31 to 46 % and 14 to 22 % at DTC and UZF, respectively. However, SOC concentration in NTM was equal ($p > 0.05$) to NTR and NTMR treatments at both sites.

In the 5-10 cm soil layer of DTC, SOC concentrations in NTM and NTR were at least 19 % higher ($p = 0.005$) than in NT and CT (Table S2). There were no significant ($p > 0.05$) treatment effects on SOC concentration below 10 cm soil depth at DTC and below 5 cm depth at UZF.



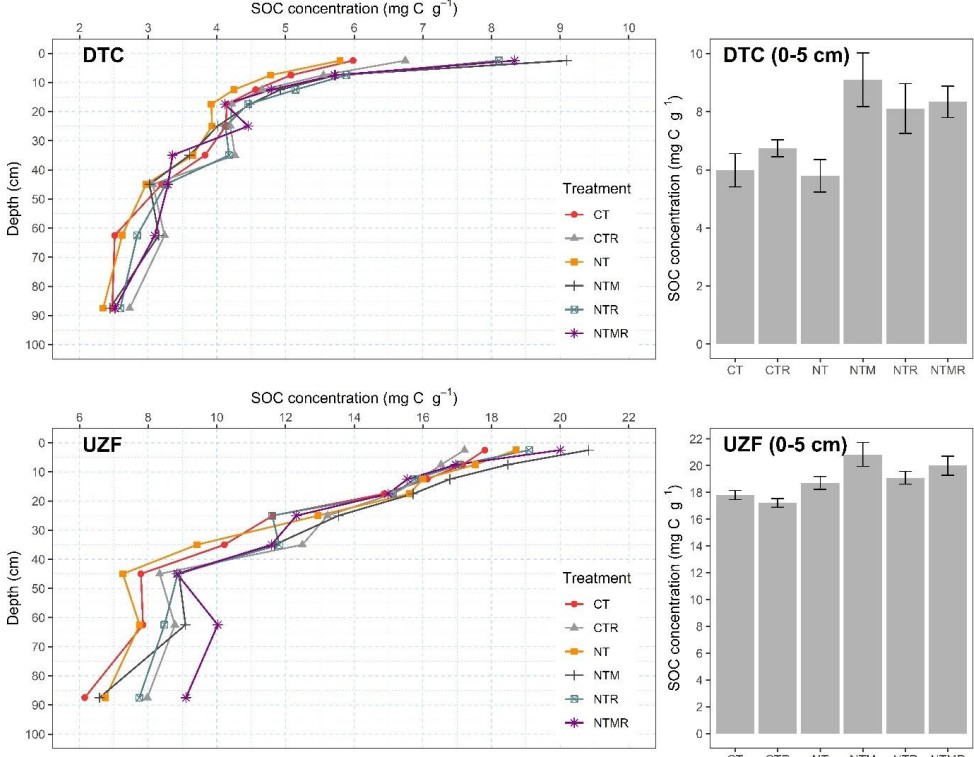

**Figure 3:** Soil depth distribution of organic carbon concentration for the different experimental treatments at the Domboshava Training Centre (DTC) and University of Zimbabwe Farm (UZF) experimental sites in Zimbabwe. Error bars represent standard errors (N = 4). CT: conventional tillage, CTR: conventional tillage with rotation, NT: no-tillage, NTM: no-tillage with mulch, NTR: no-tillage with rotation, NTMR: no-tillage with mulch and rotation.

### 3.3 SOC stocks

There were significant ($p < 0.05$) treatment effects on SOC stocks per soil layer in the 0-5 and 5-10 cm soil layers at DTC and the 0-5 cm soil layer at UZF (Table S3). Compared to CT, CTR and NT, NTM had at least 1.1 and 1.3 times more SOC stocks in the top 5 and 10 cm layers at UZF and DTC, respectively. In terms of cumulative SOC stocks, significant ($p < 0.05$)

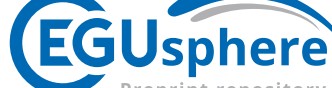

treatment effects were limited to the top 30 cm soil layer at DTC and the 20 cm layer at UZF,
where no tillage with mulching (NTM) increased SOC stocks (Table 1). There were no
significant (p > 0.05) tillage effects on SOC stocks (CT vs NT) for both sites. The rotation
component had no significant (p > 0.05) effects on SOC stocks when comparing CTR and NTR
at DTC. However, the maize-cowpea rotation under NT (NTR) had at least 16 % higher SOC
stocks in the top 30 cm compared to NT. In contrast, NTR had at least 7 % more SOC stocks
than CTR in the top 10 cm soil layer at UZF, though there were no significant (p > 0.05)
differences in SOC stocks between NTR and NT. Compared to NT and CT, the mulching
component significantly (p < 0.05) increased SOC stocks by at least 8 % at UZF and 13 % at
DTC in the top 20 and 30 cm soil layers, respectively. SOC stocks in the full CA treatment
(NTMR) were not significantly (p > 0.05) different with the other combinations of CA
principles (NTM, NTR) and CTR at DTC. At UZF, the full CA treatment had similar SOC
stocks as all the other NT treatments (NT, NTM, NTR).
SOC stocks for the whole soil profile for this study (0-100 cm) were at least 8.1, 3.5 and 2.1
times higher at DTC and 11.6, 4.4 and 2.4 times higher at UZF than the SOC stocks in the 0-5
cm (surface soil), 0-15 cm (tillage depth) and 0-30 cm (IPCC standard depth) layers. SOC
stocks for the subsoil (30-100 cm) ranged from 18.4 to 51.4 Mg C ha$^{-1}$ at DTC and 41.9 to
124.9 Mg C ha$^{-1}$ at UZF. Therefore, subsoil represented more than half (at least 53 % at DTC
and 58 % at UZF) of SOC stocks for the whole 100 cm soil profile.
**Table 1:** Cumulative SOC stocks at the Domboshava Training Centre (DTC) and University
of Zimbabwe Farm (UZF) after 8 years of different tillage, residue and crop management
systems. Means in the same row followed by different superscript letters are significantly
different and associated errors are standard errors (N = 4). CT: conventional tillage, CTR:





conventional tillage with rotation, NT: no-tillage, NTM: no-tillage with mulch, NTR: no-tillage
with rotation, NTMR: no-tillage with mulch and rotation.

| Site | Cumulative ESM (Mg ha⁻¹) | Approximate soil depth (cm) | Cumulative SOC stocks (Mg C ha-1) | | | | | | LSD | Significance |
|---|---|---|---|---|---|---|---|---|---|---|
| | | | CT | CTR | NT | NTM | NTR | NTMR | | |
| DTC | 650 | 0-5 | $3.9 \pm 0.8^{c}$ | $4.4 \pm 0.4^{bc}$ | $3.8 \pm 0.7^{c}$ | $5.9 \pm 1.2^{a}$ | $5.3 \pm 1.1^{ab}$ | $5.4 \pm 0.7^{ab}$ | 0.9 | $p < 0.001$ |
| | 1340 | 0-10 | $7.4 \pm 1.3^{c}$ | $8.2 \pm 0.6^{bc}$ | $7.1 \pm 1.1^{c}$ | $9.9 \pm 1.8^{a}$ | $9.4 \pm 1.5^{ab}$ | $9.4 \pm 1.1^{ab}$ | 1.2 | $p < 0.001$ |
| | 2060 | 0-15 | $10.7 \pm 1.6^{c}$ | $11.6 \pm 0.8^{bc}$ | $10.1 \pm 1.3^{c}$ | $13.5 \pm 2.0^{a}$ | $13.1 \pm 1.7^{ab}$ | $12.9 \pm 1.2^{ab}$ | 1.7 | $p < 0.05$ |
| | 2760 | 0-20 | $13.6 \pm 1.7^{b}$ | $14.6 \pm 1.0^{ab}$ | $12.9 \pm 1.4^{b}$ | $16.7 \pm 2.1^{a}$ | $16.2 \pm 1.9^{a}$ | $15.8 \pm 1.4^{a}$ | 2.1 | $p < 0.05$ |
| | 4160 | 0-30 | $19.4 \pm 1.9^{ab}$ | $20.5 \pm 1.2^{ab}$ | $18.4 \pm 1.6^{b}$ | $22.3 \pm 2.2^{a}$ | $22.0 \pm 1.9^{a}$ | $22.0 \pm 1.5^{a}$ | 2.7 | $p < 0.05$ |
| | 5590 | 0-40 | $24.9 \pm 2.0^{a}$ | $26.6 \pm 1.3^{a}$ | $23.7 \pm 1.7^{a}$ | $27.5 \pm 2.3^{a}$ | $27.9 \pm 2.0^{a}$ | $26.9 \pm 1.6^{a}$ | 3.1 | ns |
| | 7040 | 0-50 | $29.6 \pm 1.9^{a}$ | $31.2 \pm 1.3^{a}$ | $28.0 \pm 1.8^{a}$ | $32.0 \pm 2.4^{a}$ | $32.7 \pm 2.1^{a}$ | $31.7 \pm 1.7^{a}$ | 3.4 | ns |
| | 10550 | 0-75 | $38.5 \pm 2.0^{a}$ | $42.6 \pm 1.3^{a}$ | $37.3 \pm 2.0^{a}$ | $39.5 \pm 2.4^{a}$ | $42.7 \pm 2.1^{a}$ | $42.6 \pm 1.9^{a}$ | 5.2 | ns |
| | 13770 | 0-100 | $46.5 \pm 2.0^{a}$ | $51.4 \pm 1.3^{a}$ | $44.8 \pm 2.0^{a}$ | $47.5 \pm 2.4^{a}$ | $51.1 \pm 2.2^{a}$ | $50.7 \pm 2.0^{a}$ | 6.3 | ns |
| UZF | 460 | 0-5 | $8.2 \pm 0.9^{cd}$ | $7.9 \pm 0.5^{d}$ | $8.6 \pm 0.6^{bc}$ | $9.6 \pm 1.0^{a}$ | $8.8 \pm 0.9^{bc}$ | $9.2 \pm 0.9^{ab}$ | 0.7 | $p < 0.001$ |
| | 870 | 0-10 | $15.4 \pm 1.5^{bc}$ | $14.8 \pm 1.0^{c}$ | $15.9 \pm 1.3^{b}$ | $17.3 \pm 1.7^{a}$ | $15.9 \pm 1.6^{b}$ | $16.3 \pm 1.4^{ab}$ | 1.1 | $p < 0.05$ |
| | 1330 | 0-15 | $22.9 \pm 1.9^{b}$ | $22.1 \pm 1.6^{b}$ | $23.4 \pm 1.8^{b}$ | $25.1 \pm 2.1^{a}$ | $23.2 \pm 1.9^{b}$ | $23.6 \pm 1.7^{ab}$ | 1.7 | $p < 0.05$ |
| | 1840 | 0-20 | $30.8 \pm 2.2^{b}$ | $29.9 \pm 2.1^{b}$ | $31.3 \pm 2.0^{ab}$ | $33.3 \pm 2.4^{a}$ | $30.9 \pm 2.2^{b}$ | $31.0 \pm 2.1^{b}$ | 2 | $p < 0.05$ |
| | 2760 | 0-30 | $42.3 \pm 2.4^{a}$ | $42.8 \pm 2.2^{a}$ | $44.1 \pm 2.1^{a}$ | $46.4 \pm 2.8^{a}$ | $41.9 \pm 2.7^{a}$ | $43.3 \pm 2.7^{a}$ | 3.3 | ns |
| | 4030 | 0-40 | $55.2 \pm 2.6^{a}$ | $58.1 \pm 2.6^{a}$ | $57.2 \pm 2.2^{a}$ | $61.0 \pm 3.3^{a}$ | $56.7 \pm 3.0^{a}$ | $57.5 \pm 3.2^{a}$ | 4.8 | ns |
| | 5300 | 0-50 | $66.3 \pm 2.7^{a}$ | $70.4 \pm 3.0^{a}$ | $67.5 \pm 2.3^{a}$ | $73.1 \pm 3.9^{a}$ | $68.8 \pm 3.1^{a}$ | $69.7 \pm 3.3^{a}$ | 6.6 | ns |
| | 8190 | 0-75 | $89.3 \pm 3.1^{a}$ | $95.9 \pm 3.3^{a}$ | $90.0 \pm 2.7^{a}$ | $89.9 \pm 4.6^{a}$ | $93.7 \pm 3.9^{a}$ | $98.4 \pm 4.3^{a}$ | 17 | ns |
| | 11050 | 0-100 | $107.8 \pm 3.5^{a}$ | $119.1 \pm 3.7^{a}$ | $109.8 \pm 3.3^{a}$ | $110.9 \pm 5.2^{a}$ | $116.1 \pm 4.9^{a}$ | $124.9 \pm 5.6^{a}$ | 19 | ns |



### 3.4 SOC accumulation and loss rates

SOC accumulation rates differed significantly ($p < 0.05$) with soil depth where top soil layers (0-5, 0-10, 0-15, 0-20 and 0-30 cm) had SOC accumulation rates that were at least 6.9 times less than when considering the 0-100 cm soil profile at UZF (Table 2). In contrast, there were no significant ($p > 0.05$) differences in SOC accumulation rates with depth at DTC. On average, SOC accumulation rates ranged between 0.13 Mg C ha$^{-1}$ yr$^{-1}$ in the top soil (0-5 cm) to 0.33 Mg C ha$^{-1}$ yr$^{-1}$ for the whole 1 m soil profile at DTC. The depth and treatment interaction had no significant ($p > 0.05$) effects at both sites.

On the other hand, the different treatments in this study had significant ($p < 0.05$) effects in SOC accumulation rates in the top 20 cm soil layer at both sites (Table 2). At DTC, NT had significant ($p < 0.05$) net loss of SOC in the 0-20 cm layer, ranging between -0.09 and -0.02 Mg C ha$^{-1}$ yr$^{-1}$, whereas the treatments with different combinations of CA principles under NT (NTM, NTR, NTMR) has accumulation rates ranging from 0.17 to 0.38 Mg C ha$^{-1}$ yr$^{-1}$. However, maize stover mulching (NTM) had significantly ($p < 0.05$) higher SOC accumulation rates than CTR (2.9 – 4.2 times) and NT (5.2 – 13.5 times) in the top 15 cm and 20 cm layers, respectively. The different combinations of mulching and rotation under NT had no significant ($p > 0.05$) differences in SOC accumulation rates. Similarly, rotation treatments (CTR, NTR, NTMR) showed no significant ($p > 0.05$) differences in SOC accumulation rates. Thus, the full CA treatment had similar SOC accumulations rates to treatments with at least 2 combinations of CA principles (NTM and NTR) and to CTR.

In contrast, at UZF, CTR had significant ($p < 0.05$) net loss of SOC in the top 20 cm (Table 2). The no-tillage treatments (NT, NTM, NTR, NTMR) showed significantly ($p < 0.05$) higher SOC accumulation rates (0.05 – 0.25 Mg C ha$^{-1}$ yr$^{-1}$) than CTR which ranged between -0.07 to -0.03 Mg C ha$^{-1}$ yr$^{-1}$ in top 10 cm soil layer. NTM had the highest SOC accumulations rates



(0.28 to 0.32 Mg C ha$^{-1}$ yr$^{-1}$) when considering the 0-15 and 0-20 cm soil layers. SOC
accumulation rates in NTM were at least 3.8, 3.5 and 2.4 times higher than CTR, NT and NTR
in the top 20 cm. The full CA treatment (NTMR) had significantly ($p < 0.05$) higher SOC
accumulation rates compared to CTR (2.5 – 5.3 times) in the top 10 cm and lower SOC
accumulation rate to NTM in the 0-10 cm (2.3 times) and 0-20 cm (10.6 times) soil layer.
However, there were no significant ($p > 0.05$) differences in SOC accumulation rates between
treatments beyond 20 cm soil layer at both sites.
**Table 2:** SOC change rates (± standard error, N = 4) of the different treatments compared to
CT (conventional tillage) at Domboshava Training Centre (DTC) and University of Zimbabwe
farm (UZF). Means in the same row followed by different superscripts are significantly
different. CTR: conventional tillage with rotation, NT: no-tillage, NTM: no-tillage with mulch,
NTR: no-tillage with rotation, NTMR: no-tillage with mulch and rotation, LSD = least
significance difference, ns = not significant, Sig = significance, ** = $p < 0.05$, *** = $p < 0.001$.

| Site | Approximate soil depth (cm) | SOC accumulation or loss rate (Mg C ha$^{-1}$ yr$^{-1}$) | | | | | LSD | Sig |
|------|------|------|------|------|------|------|------|------|
| | | CTR | NT | NTM | NTR | NTMR | | |
| | 0-5 | 0.06 ± 0.05[bc] | -0.02 ± 0.02[c] | 0.25 ± 0.05[a] | 0.17 ± 0.02[ab] | 0.19 ± 0.04[ab] | 0.13 | *** |
| | 0-10 | 0.10 ± 0.09[bc] | -0.04 ± 0.04[c] | 0.31 ± 0.09[a] | 0.24 ± 0.01[ab] | 0.25 ± 0.08[ab] | 0.16 | *** |
| | 0-15 | 0.12 ± 0.13[bc] | -0.07 ± 0.05[c] | 0.35 ± 0.13[a] | 0.30 ± 0.01[ab] | 0.27 ± 0.12[ab] | 0.23 | ** |
| | 0-20 | 0.12 ± 0.17[ab] | -0.09 ± 0.05[b] | 0.38 ± 0.17[a] | 0.32 ± 0.01[a] | 0.27 ± 0.16[a] | 0.29 | ** |
| DTC | 0-30 | 0.13 ± 0.25[a] | -0.13 ± 0.07[a] | 0.36 ± 0.25[a] | 0.32 ± 0.08[a] | 0.33 ± 0.20[a] | 0.35 | ns |
| | 0-40 | 0.22 ± 0.25[a] | -0.15 ± 0.07[a] | 0.33 ± 0.25[a] | 0.38 ± 0.07[a] | 0.25 ± 0.23[a] | 0.41 | ns |
| | 0-50 | 0.20 ± 0.27[a] | -0.20 ± 0.14[a] | 0.30 ± 0.27[a] | 0.40 ± 0.09[a] | 0.26 ± 0.22[a] | 0.46 | ns |
| | 0-75 | 0.51 ± 0.28[a] | -0.15 ± 0.28[a] | 0.13 ± 0.28[a] | 0.53 ± 0.13[a] | 0.51 ± 0.20[a] | 0.73 | ns |
| | 0-100 | 0.62 ± 0.32[a] | -0.20 ± 0.37[a] | 0.13 ± 0.32[a] | 0.58 ± 0.29[a] | 0.53 ± 0.20[a] | 0.86 | ns |
| | 0-5 | -0.03 ± 0.03[c] | 0.05 ± 0.04[b] | 0.17 ± 0.05[a] | 0.07 ± 0.04[b] | 0.13 ± 0.06[ab] | 0.08 | *** |
| | 0-10 | -0.07 ± 0.04[c] | 0.07 ± 0.08[b] | 0.25 ± 0.09[a] | 0.07 ± 0.07[b] | 0.11 ± 0.08[b] | 0.13 | ** |
| | 0-15 | -0.10 ± 0.03[b] | 0.06 ± 0.11[b] | 0.28 ± 0.13[a] | 0.04 ± 0.07[b] | 0.09 ± 0.11[ab] | 0.22 | ** |
| | 0-20 | -0.11 ± 0.07[b] | 0.06 ± 0.14[b] | 0.32 ± 0.17[a] | 0.02 ± 0.11[b] | 0.03 ± 0.12[b] | 0.25 | ** |
| UZF | 0-30 | 0.06 ± 0.15[a] | 0.22 ± 0.25[a] | 0.51 ± 0.25[a] | -0.05 ± 0.18[a] | 0.12 ± 0.16[a] | 0.44 | ns |
| | 0-40 | 0.37 ± 0.11[a] | 0.25 ± 0.27[a] | 0.72 ± 0.25[a] | 0.19 ± 0.28[a] | 0.29 ± 0.14[a] | 0.65 | ns |
| | 0-50 | 0.51 ± 0.20[a] | 0.15 ± 0.34[a] | 0.85 ± 0.27[a] | 0.31 ± 0.41[a] | 0.43 ± 0.10[a] | 0.88 | ns |
| | 0-75 | 0.83 ± 0.56[a] | 0.08 ± 0.55[a] | 0.08 ± 0.28[a] | 0.55 ± 0.76[a] | 1.14 ± 0.44[a] | 1.17 | ns |
| | 0-100 | 1.41 ± 0.86[a] | 0.25 ± 0.75[a] | 0.98 ± 0.32[a] | 1.03 ± 1.26[a] | 2.14 ± 0.99[a] | 2.31 | ns |





**3.5 Organic carbon inputs via crops residues, root mortality and rhizodeposition to the soil**

Figure 4 presents cumulative OC inputs which significantly ($p < 0.001$) differed between cropping systems. Cumulative OC inputs were at least 1.5 times higher in mulch treatments (NTM, NTMR) than in treatments without mulch. However, the mulch plus rotation treatment (NTMR) had significantly ($p < 0.001$) lower cumulative OC inputs than continuous mulching (NTM). Cumulative OC input after 8 seasons was as high as 16.0 and 10.5 Mg C ha$^{-1}$ at DTC and 16.2 and 12.4 Mg C ha$^{-1}$ at UZF in NTM and NTMR, respectively (Figure 4), resulting in mean seasonal OC input rates of about 1.3 to 1.6 Mg C ha$^{-1}$ season$^{-1}$ for NTMR and 2.0 Mg C ha$^{-1}$ season$^{-1}$ for NTM. The other treatments had mean seasonal OC input rates $\leq 1.0$ Mg C ha$^{-1}$ season$^{-1}$.

**3.6 Soil $CO_2$-C efflux and cumulative emissions**

Daily soil $CO_2$ fluxes were significantly ($p < 0.05$) higher in the maize rows than the inter-rows at both sites (Figure 5 and S2). However, there were no significant ($p > 005$) differences in daily $CO_2$ fluxes between treatments. Fluxes of $CO_2$ spiked at maximum maize vegetative stage (from approximately 25 to 100 days after germination) in the rainy season and tailed off to $< 50$ mg $CO_2$-C m$^{-2}$ hr$^{-1}$ after harvesting and into the dry season (May to September 2021).

There were no significant ($p > 0.05$) differences in cumulative $CO_2$-C emissions for both seasons and sites (Figure 6). Cumulative $CO_2$-C emissions ranged from 5.0 to 6.2 Mg $CO_2$-C ha$^{-1}$ yr$^{-1}$ and 5.9 to 7.5 Mg $CO_2$-C ha$^{-1}$ yr$^{-1}$ at DTC and UZF, respectively, in the 2019/20 cropping season. In the 2020/21 season, cumulative $CO_2$-C emissions ranged between 4.3 to 6.3 Mg $CO_2$-C ha$^{-1}$ yr$^{-1}$ and 5.8 to 7.5 Mg $CO_2$-C ha$^{-1}$ yr$^{-1}$ at DTC and UZF, respectively.





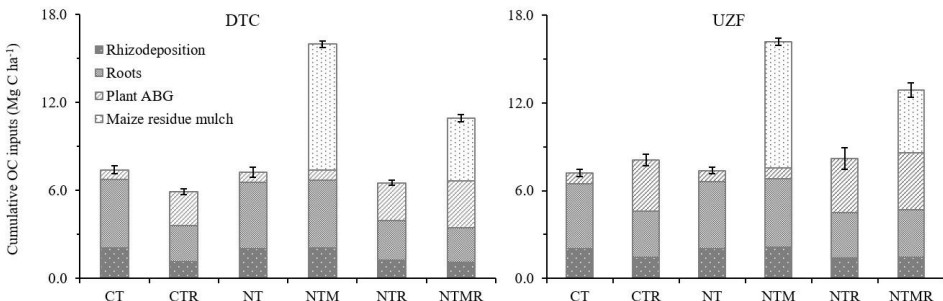

**Figure 4:** Estimated cumulative organic carbon (OC) inputs to the soil from the 2013/14 to the

2020/21 cropping season for the different treatments at the Domboshava Training Centre (DTC)

and the University of Zimbabwe Farm (UZF) experimental sites. Error bars represent standard

errors (n = 4) for the cumulative OC. CT: conventional tillage, CTR: conventional tillage with

rotation, NT: no-tillage, NTM: no-tillage with mulch, NTR: no-tillage with rotation, NTMR:

no-tillage with mulch and rotation, ABG: aboveground biomass.



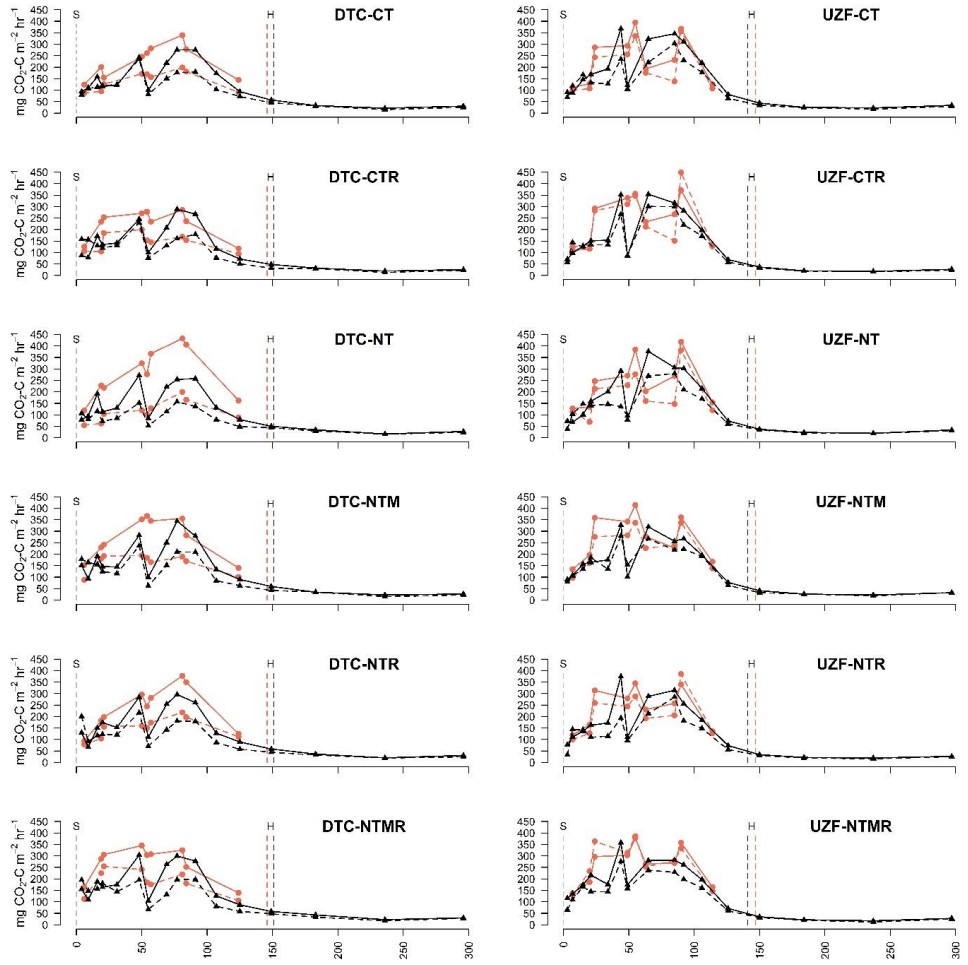

**Figure 5:** Daily $CO_2$-C fluxes during the 2019/2020 (orange) and 2020/21 (black) seasons for the different treatments at Domboshava Training Centre (DTC) and University of Zimbabwe Farm (UZF) experimental sites. Solid and dotted lines represent fluxes measured in the maize intra-row and inter-row spaces respectively; CT: conventional tillage, CTR: conventional tillage with rotation, NT: no-tillage, NTM: no-tillage with mulch, NTR: no-tillage with rotation, NTMR: no-tillage with mulch and rotation, S = sowing and H = harvesting.





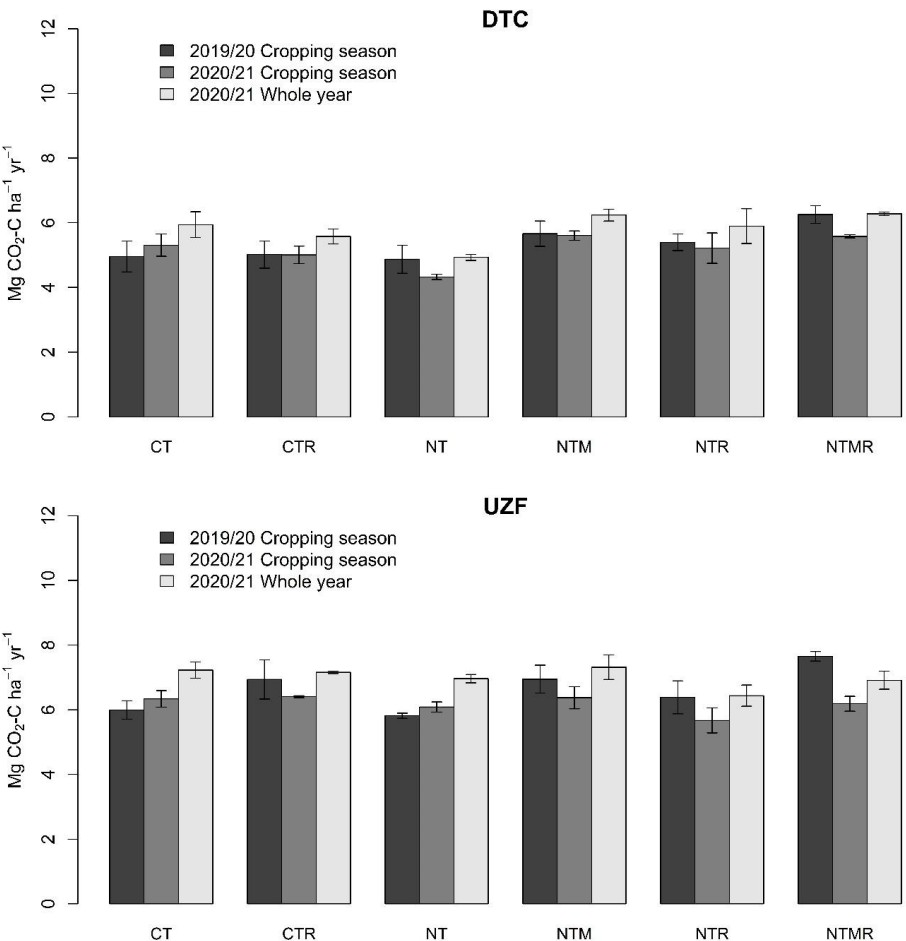

**Figure 6:** Cumulative total $CO_2$-C emissions for the different treatments in the 2019/20 cropping season, the 2020/21 cropping season, and the whole year (2020/21) at Domboshava Training Centre (DTC) and University of Zimbabwe Farm (UZF). Bars represent standard errors (N = 4). CT: conventional tillage, CTR: conventional tillage with rotation, NT: no-tillage, NTM: no-tillage with mulch, NTR: no-tillage with rotation, NTMR: no-tillage with mulch and rotation.



## 4. Discussion

### 4.1 SOC distribution across soil depth

Cumulative SOC stocks for the whole soil profile (0-100 cm) for this study were at least 8.0,
4.0 and 2.0 times higher than the 0-5 (surface soil), 0-15 (tillage depth for the study) and 0-30
(IPCC standard depth for SOC studies), for the two sites, respectively. This means that, > 50
% SOC stocks for this study were in the sub-soil (30-100 cm) which reflects on the importance
of sub-soil SOC stocks. Significant SOC stocks in the sub-soil have also been reported by other
authors (Balesdent et al., 2018; Cardinael et al., 2015; Harrison et al., 2011; Lal, 2018; Yost &
Hartemink, 2020). Significant treatment effects were restricted to the top 30 cm in our study as
well as other studies in SSA (Dube et al., 2012; Powlson et al., 2016) and the world at large
(Balesdent et al., 2018; Yost & Hartemink, 2020), which is most likely why the default soil
depth for IPCC for SOC studies is 0-30 cm (IPCC, 2019). However, this underestimates whole
soil profile C storage (Harrison et al., 2011; Lorenz & Lal, 2005; Singh et al., 2018) and hence
the need to consider depth differentiated assessments of whole soil profiles when monitoring
SOC changes in agricultural ecosystems if long term SOC storage is to be effective in the
pursuit of climate change mitigation (Malepfane et al., 2022). The differentiated soil depth
assessments of SOC also show soil depth sections that are sensitive to disturbance (tillage) and
OC inputs through above-, below-ground biomass and organic soil fertility amendments like
manure and compost. SOC mineralization is relatively low in the sub-soil due to lack of oxygen
and physical protection of SOC (aggregate protected C) (Button et al., 2022; Rumpel et al.,
2012; Sanaullah et al., 2016; Shumba et al., 2020). Therefore, crop varieties with deep rooting
systems are encouraged to be developed in the pursuit of increasing subsoil OC inputs through
root mortality and exudates.





## 4.2 Cumulative SOC stocks and accumulation rates

Significant changes in SOC stocks and accumulation rates were restricted to the top 20 cm at both sites below which there were no differences between treatments (Table 2). The consistently low SOC stocks under NT, CT and CTR and SOC accumulation rates (CTR and NT) at both sites was attributed to low OC inputs (Table S4, Figure 4). NT and CT had generally low yields in terms of grain and vegetative aboveground biomass (Mhlanga et al., 2021; Shumba et al., 2022) since the establishment of the experiment in the 2013/14 season hence low OC inputs through stubble, root mortality and root exudates. Our results dovetails with studies done elsewhere (Abdalla et al., 2016; Du et al., 2017; Koga & Tsuji, 2009) and meta-analyses and reviews (Corbeels, Cardinael, et al., 2020; Lal, 2015, 2018) where the authors found that NT alone does not significantly improve SOC. However, higher SOC stocks were observed when NT was combined with at least two CA principles (mulching and rotation) at DTC in the top 20 cm (Table 1). It has been reported that NT cropping systems does not necessarily add SOC but their contribution to SOC accumulation is largely accomplished by increasing C inputs in the top layers and reducing erosion through minimum soil disturbance (Bai et al., 2019; Lal, 2015, 2018; Six et al., 2000). Thus, NT without mulch is a nonentity compared to other combinations of CA principles for long-term sustainability in cropping systems (Bohoussou et al., 2022; Kodzwa et al., 2020; Li et al., 2020; Mhlanga et al., 2021; Nyamangara et al., 2013) and NT is only effective in increasing SOC stocks when it is associated with other CA principles, especially mulch.

On the other hand, our study suggest that NT can achieve the same results of SOC storage as NTR and NTMR since they had similar SOC stocks at UZF. This can be explained by the low aboveground OC inputs in rotation treatments during the season when cowpeas were grown. Therefore, the differences in the response to SOC changes and accumulation rates to the



treatments in this study, where NT and CTR had the lowest SOC accumulation rates at DTC
and UZF respectively, suggests that SOC storage is, as expected, site specific.
Legume rotations have been found to improve SOC accumulation rates and subsequent soil
structural improvement (aggregation) induced by the addition of organic residues with
favourable C/N ratio (Jephita et al., 2023; Laub et al., 2023; Virk et al., 2022). However, in our
study, cowpea rotation benefits on SOC accumulation rates were not observed in comparison
to monocropping under NTM at DTC. Nevertheless, benefits from cowpea rotation under NT
cropping systems (NTR, NTMR) compared to CT cropping systems (CTR) were significant,
albeit only in the top 10 cm, at UZF; CTR had a net loss of SOC (-0.07 ± 0.04 to 0.03 ± 0.03
Mg C ha$^{-1}$ yr$^{-1}$). In addition to low OC inputs (Figure 4), the net SOC loss in CTR was due to
seasonal exposure to oxidative losses (SOC mineralization) through disruption of soil
macroaggregates by tillage as alluded by Bai et al., (2019); Cambardella and Elliott, (1993)
and Lal, (2018).
Overall, the insignificant differences in cumulative SOC stocks and SOC accumulation / loss
rates between the different cropping systems in this and other studies (Laura et al., 2022; Leal
et al., 2020) when considering whole soil profile (0-100 cm) is alluded to the dilution effect
due to low OC inputs beyond 30 cm. Organic C inputs in the subsoil (> 30 cm) through crop
root mortality and root exudates are highly reduced due to low root biomass (Button et al.,
2022; Chikowo et al., 2003). Several authors have also reported similar cumulative SOC stocks
for soil profile depth > 30 cm between different tillage and residue management practices
(Angers et al., 1997; Doran et al., 1998; Lal, 2015; Powlson et al., 2014). This can be alluded
to an accumulation of uncertainty when cumulating SOC stocks in several soil layers with their
respective error of measurement, which complicates the task of detecting statistically
significant differences even where such differences exist (Kravchenko & Robertson, 2011).



Kravchenko and Robertson, (2011) bemoaned the lack of enough replication when sampling
deep soil horizons to reduce variability and the importance of post hoc power analysis to reduce
Type II error.

**4.3 Role of soil texture in SOC accumulation**
Soil texture is widely recognized to influence SOC stocks (Sun et al., 2020) through physical
and chemical protection of SOC against microbially mediated decomposition (Chivenge et al.,
2007; Mtambanengwe et al., 2004). In our study, the main difference between the two study
sites is soil texture in the top soil (0-30 cm), where DTC had light textured (sandy loams) soils
and UZF had medium to heavy textured (sandy-clay-loams) soils (Figure 1). These soil textural
differences explain why there were no differences in SOC stocks, changes and accumulation
rates between NTM, NTR and NTMR at DTC regardless of higher cumulative OC inputs in
NTM and NTMR (Figure 4). The direct SOC inputs in the top soil, where SOC was more
concentrated (Table S2, Figure 3) was subject to mineralization because of low clay content
and thus low protection by soil micro-aggregates (Chivenge et al., 2007; Mtambanengwe et al.,
2004; Sun et al., 2020), such that the differences in OC inputs had little effect. Light textured
soils have large pores which cannot protect SOC against microbial decomposition
(Christensen, 1987; Kravchenko & Guber, 2017; Mtambanengwe et al., 2004; Sun et al., 2020).
In contrast, there were differences between NTM and NTR at UZF in the top soil layers and
intermediate between NTM and NTMR. Cumulative OC inputs in NTMR (12.4 Mg C ha$^{-1}$)
were about 75 % of cumulative OC inputs in NTM (16.2 Mg C ha$^{-1}$) (Figure 4) after 8 seasons.
The added C, especially from maize stover mulch, most likely was protected by clay particles
as well as formation of organo-mineral complexes (Chivenge et al., 2007; Jephita et al., 2023;



Malepfane et al., 2022) which protects SOC from mineralization (Button et al., 2022; Dunjana
et al., 2012; Rumpel et al., 2012; Sanaullah et al., 2016; Shumba et al., 2020).

**4.4 Soil $CO_2$ fluxes and cumulative $CO_2$ emissions**
Despite the higher OC inputs in NT cropping systems with mulch, $CO_2$ fluxes and cumulative
emissions were similar between treatments. This is attributed to the fact that the $CO_2$ fluxes in
this study were the sum of autotrophic and heterotrophic respiration (Heinemeyer et al., 2007);
hence possible treatment effects on heterotrophic respiration were most likely masked. Maize
root respiration (autotrophic respiration) has been shown to contribute an average of about 45
% (Hao and Jiang, 2014) to total soil respiration (heterotrophic and autotrophic respiration). In
contrast to other studies (Carbonell-Bojollo et al., 2019; Chatskikh and Olesen, 2007;
McDonald et al., 2019; O'Dell et al., 2020), no higher fluxes and emissions were observed
following top soil disturbance in the CT treatments (Figures 5 and 6). This was attributed to
low SOC stocks in CT treatments in the top 15 cm which was the plough depth in this study.

**5. Conclusions**
Our study has shown the overarching importance of combining at least two CA principles to
improve top SOC stocks. Mulching under no tillage system (NTM) improves SOC stocks in
the top soil though the same can be achieved by the full CA (NTMR), or no tillage plus rotation
(NTR) cropping system on a sandy soil. The absence of tillage alone (NT) could not increase
SOC stocks, and even lead to a slight decrease compared to CT, due to lower crop productivity
in NT and therefore reduced OC inputs to the soil. Nevertheless, whole profile (0-100 cm) SOC
stocks was the same between all the treatment. Our study also showed that sampling the entire



soil profile is necessary for a more accurate view of SOC accumulation potential among
different cropping systems.

## 6. Author contributions

CT designed, established and maintained the experiments since 2013,
AS, RCa, RCh were involved in various gas and soil sampling campaigns; JS was involved in
laboratory analysis of gas samples,
AS, RCa performed the statistical analyses, graphics and drafting the manuscript,
AS, RCa, RCh, MC, JS, CT reviewed and edited the manuscript

## 7. Competing interests

One of the co-authors is a member of the editorial board of the SOIL journal. The other authors
have no competing interests to declare.

## 8. Acknowledgements

This study was funded by the DSCATT project "Agricultural Intensification and Dynamics of
Soil Carbon Sequestration in Tropical and Temperate Farming Systems" (N∘ AF 1802-001, N∘
FT C002181), supported by the Agropolis Foundation ("Programme d'Investissement
d'Avenir" Labex Agro, ANR-10-LABX- 0001-01) and by the TOTAL Foundation within a
patronage agreement. Authors are grateful to the International Maize and Wheat Improvement
Center (CIMMYT) for the setup and running of the experiment. We also acknowledge the
donors of the MAIZE CGIAR Research Program (www.maize.org) and the Ukama Ustawi



Regional CGIAR Initiative who supported he trials up to 2018 and staff time until 2023. We
thank Britta Jahn-Humphrey for carrying out the gas analyses at ETH Zürich. We also thank
Admire Muwati for his help in gas sampling. Special thanks go to the technical personnel at
each experimental locations namely Tarirai Muoni, Sign Phiri, Herbert Chipara and Connie
Madembo who continuously assisted in trial establishment and management.

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

937  04550-z

938