# Peer review of "Mulch application as the overarching factor explaining increase in soil organic carbon"

_EGUsphere, 2023_

## Author Comment (AC2)

RC2

An overall relevant contribution on use of conservation agriculture practices to improve maize crop productivity and raise carbon stocks (and in turn fertility) in Zimbabwe. The study relies on two well-established field trials and standard yet well executed assessments were used to reveal how tillage, mulching and crop rotation in isolation or jointly influence SOC in these understudied systems. While the approach is overall robust – aside from limited attention to crop C inputs – at times the interpretation remains rather superficial. I have listed a number of comments below:

We thank the Reviewer for the overall positive comment that our study was well executed scientifically and can contribute to the body of knowledge on conservation agriculture. We are grateful for the insightful comments that have improved our manuscript. Questions or comments raised by the Reviewer have been answered below. Therefore, in black are the Reviewer's comments/questions and in blue are our responses. We also answered the detailed/minor comments below.

L103 & 108 One aspect of the hypotheses is best elaborated a bit more: while it is clear that crop residue retention and adoption of no-tillage are likely to improve SOC stocks, this is less obvious so for crop rotation. Crop rotation as such rather seems like a means to combat pests, but why should there necessarily be any beneficial effect onto SOC? Indirectly perhaps, via enhanced crop growth?

We agree with the reviewer's comments. Crop rotation is indeed crucial to reduce pests and diseases. The hypothesis is that the productivity of the crops is enhanced due to this reduction is biotic pressure, and therefore carbon inputs to the soil might be increased too. The second point is that this rotation introduces a nitrogen fixing crop, cowpea. We expect the following maize crop might benefit from this additional nitrogen to the soil. The third hypothesis is that crop diversification enhances soil biological processes via different root systems, and possibly enhanced microfauna diversity and/or abundance (e.g. mycorrhizae) that could improve aggregate stability and therefore physical protection of soil carbon. The last hypothesis is that high quality residues (from the legume crop) have been shown to be preferentially stabilized in the soil due to a higher carbon use efficiency of soil microbes (Cotrufo et al., 2013; Kopittke et al., 2018). We have now better explained this in the introduction.

L300 Why carry out statistical comparisons for each individual date? This does not provide an overall view on the treatment effects and complicates drawing conclusions.

Cumulative $CO_2$ emissions per treatment and per hectare are computed using a linear interpolation between daily $CO_2$ efflux. Using sampling dates as a fixed effect in the linear mixed effect model does not prevent us from comparing emissions per treatment, it just gives an additional information on periods of the $CO_2$ effluxes. Using daily emissions in statistical models is a common practice and allows a comparison based on measurements and not on interpolation, which we think is more accurate (Kravchenko and Robertson, 2015; Barthel et al., 2022;Shumba et al., 2023).

In the UFZ site, Fig 3 reveals a rather conspicuous treatment effect of including a rotation 90cm depth (and to a lesser extent at 60cm as well). Perhaps relevant to discuss as well, even though these trends were apparently not (yet) significant. Maybe the introduction of peas particularly led to deeper root proliferation and derived C-inputs at depth… perhaps the authors could try using contrasts in the ANOVA to pool the three 'rotation' treatments vs. the other treatments and demonstrate such effect?

We concur with the observations by the Reviewer. However, there is a block effect at UZF (blocks 3 and 4) on SOC concentration in the deep soil layers as shown in Figure R1 below. We decided not to exclude the "outliers" since we thought it was a block effect rather than a treatment effect; all treatments are affected. As a result, there are no significant differences in SOC concentration between treatments in

the deep soil layers. If we exclude the "outliers" in deep soil layers the graph for SOC concentration is as shown in Figure R2. All raw data of this paper can be freely accessed on the CIRAD database and linked to this paper (https://doi.org/10.18167/DVN1/VPOCHN). We have included the link to the raw data in the materials and methods section (Section 2) and in the Data analysis sub-section (Section 2.8).

We have now also added these Figures R1 and R2 in supplementary materials and discussed implications in the discussion. At this stage we have excluded the addition of error bars on Fig 3 as the graphs will be too congested.

[Figure]

**Figure R1:** SOC concentration for the 100 cm soil profile for each treatment and replicate (block) showing "outliers" mainly in the 3rd and 4th replicates (blocks) at UZF.

[Figure]

**Figure R2:** Mean SOC concentration for the 100 cm soil profile for each treatment excluding "outliers".

The discussion section requires improvement:

Thank you for the comment and we have tackled the respective comments below.

The second half of 4.1 does not make much of a connection with own observations (L476-483) – it is not clear what message the authors wanted to bring here.

We thank the Reviewer for the insightful comment and we have rephrased the section to read as follows:

"SOC mineralization is relatively low in the subsoil due to lack of oxygen and physical protection of SOC (aggregate protected C) (Rumpel et al., 2012; Sanaullah et al., 2016; Shumba et al., 2020; Button et al., 2022). Nevertheless, belowground OC inputs are concentrated in the top 30 cm hence the recommendation for crop varieties with deep rooting systems in the pursuit of increasing subsoil OC inputs through root mortality and exudates."

However, in the section referenced (Section 4.1), we were discussing the implications of our study findings as well as suggesting alternative ways of increasing subsoil SOC. The main message we were putting across is the importance of sub-soils in SOC storage as well as stressing whole soil profile sampling in SOC monitoring studies.

Section 4.2 is lengthy - Perhaps this section could benefit from further subdivision into subparts that deal with the various treatment aspects (tillage, mulching, crop rotation).

We agree with the Reviewer's comments and we have subdivided section 4.2 accordingly into three sub-sections which dealt with the different treatment aspects in our study. Sub-section 4.2.1, 4.2.2 and 4.2.3 dealt with mulching, tillage and maize-cowpea rotation aspects, respectively. Some editorials and references were also added to improve the section.

4.2 lacks a clear calculation of conversion efficiency of C-inputs into SOC. There have been ample studies on removal or incorporation of maize residues (in light of research into the value of root vs. shoot C inputs) – such studies have generally revealed that aboveground biomass C inputs are far less effective in sustaining SOC – and this body of literature should be used to complement the discussion here. Even though much of the C-inputs are just derived from yield data, it would still be relevant to see some estimates of the efficiency by which these C-inputs formed or sustained SOC.

Thank you for the suggestion to calculate conversion efficiency of C-inputs into SOC. We tried to calculate this efficiency in the 0-30 cm depth for some treatments only. It was indeed not possible to calculate it for treatments with lower SOC stocks than the reference, or for treatments having higher SOC stocks but with less C-inputs than the reference (for example NTR at DTC). Here is the result:

| Treatments | DTC | UZF |
| --- | --- | --- |
| NTM vs CT | 0.34 | 0.46 |
| NTM vs NT | 0.45 | 0.27 |
| NTMR vs CT | 0.75 | 0.17 |
| NTMR vs NT | 0.98 | |

Some of these rates are unrealistic, especially the rates of NTMR at DTC. This is due to the high uncertainty linked to the estimation of C inputs to the soil, but also to the low number of replicates (4) for SOC stocks. Another explanation could be that most additional SOC might be in the form of labile particulate organic matter < 2mm, i.e decomposing plant materials accumulating in the soil, but not really stabilized carbon. While the calculation of the conversion efficiency of C-inputs into SOC could have been of interest, we believe they are too uncertain to show them. We have therefore decided not to include them in the manuscript.

We also took cognisant of the comments raised by the Reviewer on aboveground- vs belowground biomass in sustaining SOC. We have discussed the essence of belowground biomass to SOC storage in Section 4.2 (sub-section 4.2.1) and giving relevant literature (Hirte et al., 2021, 2018; Jones et al., 2009; Villarino et al., 2021) in support of the discussion.

The discussion lacks considerations on the potential mechanisms through which the inclusion of peas in the crop rotation might influence maize C inputs. With a legume introduced there may well be more N-available for the maize crop – not trivial given the very low doses of mineral N-supplied. If so, we may expect this to hold an effect on the soil C balance not only through an overall stimulation of maize productivity (and that is currently not clearly explained in the current discussion) but possibly also by effects onto the maize rooting pattern. These aspects should at least be commented upon, especially in light of the rather conspicuous difference in SOC at 75 or 90cm depth when a rotation is included

(at least such would seem so from Fig 3, but unfortunately no comparison is given of the 75-100cm SOC stocks).

In concurring with the Reviewer, we have included the nitrogen input through biological nitrogen fixation from cowpeas in stimulating productivity of the succeeding maize crop. We also included another potential SOC stabilization mechanism in the form of addition of high-quality cowpea residues (above-and belowground biomass) which have been shown to be preferentially stabilized in the soil due to a higher carbon use efficiency of soil microbes (Cotrufo et al., 2013; Kopittke et al., 2018).

In our case, we think the possible rotation effects in deep soil SOC stocks at UZF is an artefact due to a block effect as explained above with graphical illustrations (Figures R1 and R2). We have now also discussed that.

Towards the end of the discussion and in the conclusion clearly the specificity of the here established crop rotation needs to be stipulated. By no means could we simply extrapolate found results to other 'crop rotations' – with different crops and plant C-inputs.

We agree with the comment and we have improved the section where we discussed the issue of OC quality in terms of C:N ratio where there was alternate addition of low- (maize) and high- (cowpea) quality OC inputs in the cowpea-maize rotation treatment especially in sub-section 4.2.3. We also discussed the relevance of these rotation systems which had significant effects on SOC stocks under NT systems albeit under soils with higher clay content typical of UZF in our study.

L497: what about the whole idea that reduction of tillage would promote physical protection of OM inside microaggregates – should be commented on here as well.

The comment is acceptable and we have included the suggested idea in the discussion (section 4.2.2) with some references to support it. The part which was included the discussion now reads:

"Minimum soil disturbance through NT also physically protects SOC in microaggregates from exposure to oxidative losses (Shumba et al., 2020; Six et al., 2002; Dolan et al., 2006; Liang et al., 2020). However, NT without mulch is a nonentity compared to other combinations of CA principles for long-term sustainability in cropping systems (Nyamangara et al., 2013; Kodzwa et al., 2020; Mhlanga et al., 2021; Li et al., 2020; Bohoussou et al., 2022) and NT is only effective in increasing SOC stocks when it is associated with other CA principles, especially mulch. On the other hand, our study suggest that NT can achieve the same results of SOC storage as NTR and NTMR since they had similar SOC stocks at UZF. This can be explained by the low aboveground OC inputs in rotation treatments during the season when cowpeas were grown."

L510 SOC storage does not seem very site specific but rather rotation-specific + 'the differences in the response to SOC changes' makes little sense -> rephrase

We agree with the comment and we have deleted the sentence in pursuit of compressing the section.

L518 'the net SOC loss in CTR was due to seasonal exposure to oxidative losses (SOC mineralization) through disruption of soil macroaggregates by tillage' this makes little sense since these numbers are actually obtained by comparison with the CT treatment in which there is an equally intensive disruption of soil structure.

In this case we were referring to the UZF site (now indicated), for the comparison of CTR vs NTR there was significant differences in SOC accumulation / loss rates. At DTC, it was the mulching component that had significant effects on SOC accumulation / loss rates (Table 2). Below are the new phrases we used to elaborate our point.

"However, in our study, cowpea rotation benefits on SOC accumulation rates were not significant at DTC. Maize-cowpea rotation had no significant effects on maize yield (Shumba et al., 2023b; Mhlanga et al., 2021) which corresponded to low belowground biomass as well. Instead, maize stover mulching improved maize yields at DTC."

L547 what about other SOC stabilization mechanisms?

We have added SOC adsorption to clay particles as another possible SOC stabilization mechanism.

4.4 as it is stands alone and really does not bring a clear message. It may be better to integrate this section into 4.2 (which then gets even lengthier and is therefore best split).

Thank you for the suggestion but we think this is a stand-alone section which deals with $CO_2$-C emissions. However, we have now further discussed that the conventional tillage using hand-hoes in our study was mere loosening of the top 15 cm and not full soil inversion done by animal- or tractor-drawn ploughs.

The conclusion section is fine.

---

## Author Comment (AC3)

EC2

The current paper describes how conservation tillage management may or may not impact SOC stocks in two field experiments in Zimbabwe. While this theme is not very novel, the study does stand out in targeting understudied SSA-cropping systems and in the robustness of the methodological approach. It is unfortunate that only little effort was put into quantifying root biomass and overall plant-C inputs as this leaves the analysis rather speculative at points. Another critical point is that the rationale for comparing CO2 emissions between the various combinations of tillage x mulch retention x crop rotation (maize vs. maize & pea) is not really motivated. The net balancing of C-in and outputs results into changes in the SOC stock. So what is then the added value of on top comparing soil CO2 effluxes in the growing season (and once also in the 'dry season')? What did the authors hope to learn in addition to what they could not deduce from just looking at the SOC stock data? Most sections are overall in good shape, aside from the discussion. Its structure requires further work and there are a number of important comments on the at times rather limited interpretation of mechanisms behind found changes in SOC between the various treatments. If these points are adequately addressed the current paper could make a good contribution to the state-of-the-art on CA in low-input farming systems.

We would like to thank the Editor for the comments that have helped improving our manuscript. In trying to answer the comments raised above, we resorted to highlighting questions/comments raised above and answering them separately. Therefore, in black are the Editor's comments/questions and in blue are our responses. We also answered the detailed/minor comments below.

It is unfortunate that only little effort was put into quantifying root biomass and overall plant-C inputs as this leaves the analysis rather speculative at points.

It is true that we had to estimate belowground C-inputs using root:shoot ratios because root sampling is very destructive. The site is a long-term experiment but the plot sizes are small (6*12 m) especially for treatments with rotation (6*6 m). It was therefore not possible to sample for root biomass and distribution. However, annual aboveground biomass and inputs to the soil have been measured since the establishment of the experiment. In this case, a common practice is to estimate belowground carbon inputs using root:shoot ratios (Bolinder et al., 2007; Jones et al., 2009; Villarino et al., 2021; Cardinael et al., 2022). We agree there is uncertainty using this approach, but we believe this is still valuable to estimate carbon inputs to the soil between the different systems as it can explain the presence or absence of additional SOC storage.

We agree that the motivations behind the $CO_2$ measurements were not explained adequately. In a recent paper (Shumba et al 2023b), we presented $N_2O$ and $CH_4$ emissions. We decided to show the $CO_2$ emissions in the SOC related paper as it is easier to interpret changes in $CO_2$ efflux when looking at SOC stocks and at biomass production. The second point is that we compared $CO_2$ efflux on the maize rows and on maize inter-rows. Mulch was only applied in maize inter-rows, and fertilizer applied on the rows. We were expecting to observe a higher $CO_2$ efflux in inter-rows with mulch than in inter-rows without mulch, but we did not. We however found a higher $CO_2$ efflux on the rows than in the inter-rows, that we attribute to more autotrophic respiration from roots. While there were no significant differences in SOC stocks between rows and inter-rows, we showed that it was crucial to consider spatial heterogeneity (rows vs inter-rows) when investigating $CO_2$ efflux as a chamber position either on the rows or on the inter-rows might overestimate or underestimate fluxes at the plot scale. We have now better explained this in the manuscript.

Detailed/minor comment:

L96 and 67 seem to contradict each other, rephrase perhaps one

We agree with the comment and we rephrased L67 by condensing it and joining it with previous sentence to finally read as follows:

"The potential of CA to increase SOC stocks and thereby mitigate climate change has, however, been much debated (Corbeels et al., 2020a) but the general understanding is that, this potential is relatively low (Du et al., 2017; Powlson et al., 2014, 2016; Cheesman et al., 2016; Corbeels et al., 2020a)."

L91 'have' instead of 'has'

Agreed and we effected the changes as suggested.

L A motorized handheld corer was used to take >30cm soil samples: what diameter was then sampled?

The motorized corer had an inside diameter of 10 cm. We have included this dimension in the materials and methods section (Section 2.3).

L208 provide more details on the CHN analyzer

More details were added as suggested where we also summarised the procedure and gave the specifications of the CHN elemental analyser as follows:

"SOC was analysed in the ISO9001:2015-certified IRD LAMA's laboratory in Dakar by dry combustion on 100-mg aliquots of soil (ground to < 200 µm) using a CHN elemental analyser (Thermo Finnigan Flash EA1112, Milan, Italy)."

Fig2. The quality is rather low, consider something else than the current hatching

This comment is very helpful and the hatching for the 5-10 and 15-20 cm depths were changed in Figure 2.

Fig.3 I do not see the point in repeating the 0-5cm SOC concentration data here in the bar charts.

We have deleted the bar charts as we agree with the Editor's comments.

3.3 first part could be condensed further

In agreement with the comment, we have condensed the first paragraph without compromising on key results points.

Table 1 lowercase letter to designate significantly different means should not be placed in superscript

We agree with the comment and it was addressed accordingly in Table 1.

L382 'at UZF' comes in a bit late in this sentence, move forward. Same remark also for the next sentence and 'at DTC'

We concur with this comment and we moved both UZF and DTC forward so that readers can quickly comprehend the site being referred to. The sentences now read as follows:

"SOC accumulation rates at UZF differed significantly ($p < 0.05$) with soil depth where top soil layers (0-5, 0-10, 0-15, 0-20 and 0-30 cm) had SOC accumulation rates that were at least 6.9 times less than when considering the 0-100 cm soil profile (Table 2). In contrast, there were no significant ($p > 0.05$)

differences, at DTC in SOC accumulation rates with depth. On average, SOC accumulation rates ranged between 0.13 and 0.08 Mg C ha$^{-1}$ yr$^{-1}$ in the top soil (0-5 cm) to 0.33 and 1.16 Mg C ha$^{-1}$ yr$^{-1}$ for the whole 1 m soil profile at DTC and UZF, respectively. The depth and treatment interaction had no significant ($p > 0.05$) effects at both sites."

L384 so also provide likewise figures for UFZ

We agree with the comment and changes were addressed to include the average SOC accumulation rates for the top soil as well as the whole soil profile for UZF.

L391 'had' instead of 'has'; overall the sentences in this section 387-398 read strangely 'treatment x had net loss or accumulation of SOC' rephrase please in a more active form.

Comments were addressed as suggested and the paragraph now reads as follows:

"On the other hand, the different treatments in this study had significant ($p < 0.05$) effects in SOC accumulation / loss rates in the top 20 cm soil layer at both sites (Table 2). At DTC, NT had significant ($p < 0.05$) net loss of SOC in the 0-20 cm layer, ranging between -0.09 and -0.02 Mg C ha$^{-1}$ yr$^{-1}$, whereas NT treatments (NTM, NTR, NTMR) had SOC accumulation rates ranging from 0.17 to 0.38 Mg C ha$^{-1}$ yr$^{-1}$. However, maize stover mulching (NTM) had significantly ($p < 0.05$) higher SOC accumulation rates than CTR (2.9 – 4.2 times) and NT (5.2 – 13.5 times) in the top 15 cm and 20 cm layers, respectively. The different combinations of mulching and rotation under NT (NTM, NTR and NTMR) had no significant ($p > 0.05$) differences in SOC accumulation rates. Similarly, rotation treatments (CTR, NTR, NTMR) showed no significant ($p > 0.05$) differences in SOC accumulation rates. Thus, the full CA treatment had similar SOC accumulations rates to treatments with at least two combinations of CA principles (NTM and NTR) and to CTR."

Use the same manner to denote significance level in Tables 1 & 2

We agree with the comment and it was addressed accordingly in Table 2.

L418 rewrite this sentence

We agree with the comment and the sentence was rephrased to read "There were significant ($p < 0.001$) differences in cumulative OC inputs between treatments (Figure 4).".

L430 '0.05' I assume?

The observation is correct. It was a typo- error and we have corrected to "0.05".

L465 instead of > 50%, write maybe 'over half'

The suggestion is acceptable. We have addressed accordingly to read as follows:

"This means that, over half of SOC stocks for this study were in the sub-soil (30-100 cm) which reflects on the importance of sub-soil SOC stocks."

L469 'treatment effects' what sort of treatments?

The comment is noble. We have specified that we are referring to no-tillage (NT) plus mulch and/or rotation treatments. The statement now reads:

"Significant effects of mulch and/or rotation under NT were restricted to the top 30 cm in our study as well as other studies in SSA (Dube et al., 2012; Powlson et al., 2016) and the world at large (Balesdent et al., 2018; Yost and Hartemink, 2020), which is most likely why the default soil depth for IPCC for SOC studies is 0-30 cm (IPCC, 2019)."

L472 long and difficult to understand sentence

We concur with the comment and we have split the sentence into two. We further condensed the second sentence after splitting as follows:

"However, this underestimates whole soil profile C storage (Harrison et al., 2011; Lorenz and Lal, 2005; Singh et al., 2018). Therefore, it is crucial to consider whole soil profile sampling when monitoring SOC storage in agricultural ecosystems to determine their C sequestration potential in the pursuit of climate change mitigation (Malepfane et al., 2022)".

L491 add 'and' before 'hence'

We addressed the suggested comment.

L497 'do not' instead of 'does not', 'add SOC' is also not very well phrased further on in the sentence

Comment was considered and the sentence was rephrased accordingly.

L514 Be clear, supposedly you are referring to an N-fertilization effect onto maize, but that is not spelled out here. Do then also report on the found (in)differences in maize crop residue returns.

Thank you for the comment. We have clarified that there were no significant treatment effects due to maize-cowpea rotation, as expected from increased N cycling due to biological nitrogen fixation from the preceding cowpeas. Instead, at DTC, it was maize stover mulching that improved maize yields (Shumba et al., 2023) and subsequently belowground biomass.

L527-535 shorten and rewrite so that this reason for finding no difference in SOC stock when considering deeper soil profiles could potentially also be in part attributed to lack of replicates.

We took cognisance of the comment and we effected the suggested remark where we alluded that our study was limited to four replicates which might not have enough statistical power to detect significant differences between treatments. The statements are as follows:

"This can be alluded to an accumulation of uncertainty when cumulating SOC stocks in several soil layers with their respective error of measurement. This weakens the power of detecting statistically significant differences even where such differences exist (Kravchenko and Robertson, 2011). Kravchenko and Robertson, (2011) bemoaned the lack of enough replication when sampling deep soil horizons to reduce variability and the importance of post hoc power analysis to reduce Type II error. This study was limited to four replicates which might not have enough statical power to detect statistically significant differences between treatments."

L545 best to refer once more to DTC here

We addressed the suggested remark.

---

## Author Response (AR1)

We would like to thank the Topic Editor for the recent comments and for the opportunity to submit a revised version of our manuscript. In brief, we concurred with the Topic Editor's comment to omit all $CO_2$ efflux data from this manuscript. This prompted us to revise the whole manuscript including changing the title as well as all sections of the manuscript. In trying to answer all the comments raised, we resorted to respond separately, to earlier questions/comments raised by the Editor and Referees taking into consideration the most recent comments from the Topic Editor. Therefore, in black are the comments/questions raised and in blue are our responses. Line number referencing is according to the revised manuscript with tracked changes.

Topic Editor's comments

The authors accommodate most referees' & editor's requests, except for:

-There is still not a strong case for including $CO_2$ monitoring in this paper. While based on the data one could appreciate spatial heterogeneity in C-mineralization within maize-grown soil, that on its own does not add much to reaching the study's goals: viz. to know how tillage, mulching and crop rotation impact SOC stocks on the longer term. This becomes very clear from the insufficient section 4.4. Hence it will be necessary to omit all on $CO_2$ emission data & interpretation from the manuscript, particularly given the inability to split these emissions into a autotrophic and heterotrophic parts.

We appreciate the Topic Editor's comments and we have omitted all the $CO_2$ emission data which prompted the change in the title of the manuscript as well as all the other sections detailed below. We deleted Figure S2 in the Supplementary materials since it is showing cumulative $CO_2$ emission from the row and interrow spaces. Also deleted is Figure S1 in the Supplementary materials after considering that the same data was published in our earlier publication Shumba et al., (2023b) which cited instead (Line 145).

The new title for the manuscript is "Mulch application as the overarching factor explaining increase in soil organic carbon stocks under conservation agriculture in two 8-year-old experiments in Zimbabwe" (Lines 1-2).

**Abstract.**

Line 29 "and on soil $CO_2$ efflux" was deleted.

Line 39-40 "Gas samples were regularly collected using the static chamber method during the 2019/20 and 2020/21 cropping seasons and during the 2020/21 dry season." was deleted.

Line 46-48 "Regardless of larger organic carbon inputs in mulch treatments, there were no significant differences in $CO_2$ efflux between treatments, but it was higher in maize rows than in inter-rows as a result of autotrophic respiration from maize roots." was deleted.

We also added some few sentences (Lines 38-39 & 50-54), phrases (Line 40-41) and corrected tenses and verbs (Line 48-50) to augment the abstract.

We edited key word where we added "organic inputs" and removed "mulch" (Line 55-56).

-The abstract needs to provide a conclusion on the value of crop rotation for SOC management still.

The conclusion in the abstract was modified to include the comment above in lines 53-54 which reads:

"Our results also showed that the combination of at least two CA principles including mulch is required to increase SOC stocks in these low nitrogen input cropping systems."

**Introduction.**

In general, this section was restructured to omit $CO_2$ emission related literature.

**Materials and Methods**

We shifted up the second paragraph of Section 2.1 (lines 140-149).

We edited Section 2.4 (line 240) to include a statement that the mass proportion of coarse soil (> 2 mm) was not included in calculating SOC stocks.

Sections 2.6 and 2.7 (lines 281-324) were deleted to omit $CO_2$ gas sampling and cumulative emission calculation and Section 2.8 (lines 326-344) is now Section 2.6. Narratives on statistical analysis on $CO_2$ emission data was also deleted (lines 327-338).

**Results and Discussion.**

We edited Section 3.5 where we replaced "seasonal" with "annual" and "season" with "yr" (lines 479-481).

Section 3.6 which had reported $CO_2$ fluxes and emissions results was deleted (lines 484-494) together with Figures 5 and 6 (lines 502-515). This resulted in the deletion of Section 4.4 which discussed the $CO_2$ emission results.

We have seen it prudent in the discussion section to start with the role of texture in SOC accumulation such that formerly Section 4.3 is now Section 4.1. This has also changed formerly Section 4.2 to Section 4.2 and formerly Section 4.2 to be Section 4.3 (Lines 518-684).

**Conclusion**

We edited the conclusion to read (lines 721-729):

"Our study has shown the overarching importance of mulching and of combining at least two CA principles to improve top SOC stocks. No tillage (NT) alone could not increase SOC stocks, and even led to a slight decrease compared to CT, due to lower crop productivity in NT and therefore reduced OC inputs to the soil. Nevertheless, whole profile (0-100 cm) SOC stocks were the same between all the treatment. Our study also showed that sampling the entire soil profile is necessary for a more accurate view of SOC accumulation potential among different cropping systems."

-Quality of Fig 2 is below standards, especially the used hatching is problematic; Line thickness in Fig 3 needs to be increased – the same for Fig. 6 where the x-axis is not even visible in the pdf version.

We have tried our level best to improve the quality of Figures 2 (lines 357-363) and 3 (lines 381-382) as recommended. However, Figure 6 was deleted since we concurred with omitting all $CO_2$ emission data.

-A new line was entered "Nevertheless, belowground OC inputs are concentrated in the top 30 cm hence the recommendation for crop varieties with deep rooting systems in the pursuit of increasing subsoil OC inputs through root mortality and exudates." This sentence does not make much sense, rephrase.

We have rephrased the sentence to read as follows;

"Therefore, in the pursuit to improve subsoil (> 30 cm) SOC stocks through root mortality and exudates, crop varieties with higher root-length densities (Chikowo et al., 2003) in the subsoil are recommended."

These points need to be addressed still – hence major revision, if this contribution is to be considered further for publication.

We thank the Topic Editor for the comments above. We have addressed the comments raised as seen on the manuscript attached with tracked changes.

Further appreciation of the suggested changes to the manuscript:

- Thank you for particularly now better explaining the hypothesis that crop rotation could bring forth SOC-accumulation in the introduction.

We are grateful to the Reviewers and the Editor.

It is indeed good to show replicate plot data in supplementary material so that readers can themselves appreciate the apparent difference in SOC concentration at 60-90cm depth.

We appreciate the comments and we now show the replicate data for SOC concentration in the supplementary materials.

-With respect to the discussion:

Ok to split 4.2

I agree to the authors' decision not to add in calculations on conversion efficiency and the proposed referring to studies by Hirte et al. and others suffices.

Ok with the proposed extra discussion on N-inputs via the legume crop.

Ok with additions on physical protection of POM and considerations on representation of the included 'crop rotation' effect for other crop rotations

We really appreciate the comments that acknowledge the restructuring of the discussion section of our manuscript. The comments from the Reviewers and the Editor improved the manuscript.

**Editor (EC2)**

The current paper describes how conservation tillage management may or may not impact SOC stocks in two field experiments in Zimbabwe. While this theme is not very novel, the study does stand out in targeting understudied SSA-cropping systems and in the robustness of the methodological approach. It is unfortunate that only little effort was put into quantifying root biomass and overall plant-C inputs as this leaves the analysis rather speculative at points. Another critical point is that the rationale for comparing $CO_2$ emissions between the various combinations of tillage x mulch retention x crop rotation (maize vs. maize & pea) is not really motivated. The net balancing of C-in and outputs results into changes in the SOC stock. So what is then the added value of on top comparing soil $CO_2$ effluxes in the growing season (and once also in the 'dry season')? What did the authors hope to learn in addition to what they could not deduce from just looking at the SOC stock data? Most sections are overall in good shape, aside from the discussion. Its structure requires further work and there are a number of important comments on the at times rather limited interpretation of mechanisms behind found changes in SOC between the various treatments. If these points are adequately addressed the current paper could make a good contribution to the state-of-the-art on CA in low-input farming systems.

We would like to thank the Editor for the comments that have helped improve our manuscript and we tried our level best to answered/comment the detailed/minor comments below.

It is unfortunate that only little effort was put into quantifying root biomass and overall plant-C inputs as this leaves the analysis rather speculative at points.

It is true that we had to estimate belowground C-inputs using root:shoot ratios because root sampling is very destructive. The site is a long-term experiment but the plot sizes are small (6*12 m) especially for treatments with rotation (6*6 m). It was therefore not possible to sample for root biomass and distribution. However, annual aboveground biomass and inputs to the soil have been measured since the establishment of the experiment. In this case, a common practice is to estimate belowground carbon inputs using root:shoot ratios (Bolinder et al., 2007; Jones et al., 2009; Villarino et al., 2021; Cardinael et al., 2022). We agree there is uncertainty using this approach, but we believe this is still valuable to estimate carbon inputs to the soil between the different systems as it can explain the presence or absence of additional SOC storage.

We agree that the motivations behind the $CO_2$ measurements were not explained adequately. However, we have decided to omit all $CO_2$ emission data from this manuscript and zeroing in on SOC data.

Detailed/minor comment:

L96 and 67 seem to contradict each other, rephrase perhaps one

We agree with the comment and we rephrased by condensing it and joining it with previous sentence to finally read as follows (line 71-74):

"The potential of CA to increase SOC stocks and thereby mitigate climate change has, however, been much debated (Corbeels et al., 2020a) but the general understanding is that, this potential is relatively low (Du et al., 2017; Powlson et al., 2014, 2016; Cheesman et al., 2016; Corbeels et al., 2020a)."

Additionally, L96 (now Lines 101-102) was rephrased to omit $CO_2$ effluxes to read, "However, the effects of CA on SOC dynamics has not been widely investigated in SSA.".

L91 'have' instead of 'has'

Agreed and we effected the changes as suggested (now line 97).

L A motorized handheld corer was used to take >30cm soil samples: what diameter was then sampled?

The motorized corer had an inside diameter of 10 cm. We have included this dimension in the materials and methods section (Section 2.3, line 211).

L208 provide more details on the CHN analyzer

More details were added as suggested where we also summarised the procedure and gave the specifications of the CHN elemental analyser as follows (line 234-237):

"SOC was analysed in the ISO9001:2015-certified IRD LAMA's laboratory in Dakar by dry combustion on 100-mg aliquots of soil (ground to < 200 µm) using a CHN elemental analyser (Thermo Finnigan Flash EA1112, Milan, Italy)."

Fig2. The quality is rather low, consider something else than the current hatching

We have tried our level best to improve the quality of Figure 2 (lines 357-363) by editing the current hatching to new hatching with better contrast in addition to light and dark colours.

Fig.3 I do not see the point in repeating the 0-5cm SOC concentration data here in the bar charts.

We have deleted the bar charts as we agree with the Editor's comments (lines 381-382).

3.3 first part could be condensed further

In agreement with the comment, we have condensed the first paragraph without compromising on key results points (lines 390-401).

Table 1 lowercase letter to designate significantly different means should not be placed in superscript

We agree with the comment and it was addressed accordingly in Table 1.

L382 'at UZF' comes in a bit late in this sentence, move forward. Same remark also for the next sentence and 'at DTC'

We concur with this comment and we moved both UZF (lines 426-428) and DTC (lines 429-430) forward so that readers can quickly comprehend the sites being referred to. The sentences now read as follows:

"SOC accumulation rates at UZF differed significantly ($p < 0.05$) with soil depth where top soil layers (0-5, 0-10, 0-15, 0-20 and 0-30 cm) had SOC accumulation rates that were at least 6.9 times less than when considering the 0-100 cm soil profile (Table 2). In contrast, there were no significant ($p > 0.05$) differences, at DTC in SOC accumulation rates with depth. On average, SOC accumulation rates ranged between 0.13 and 0.08 Mg C ha$^{-1}$ yr$^{-1}$ in the top soil (0-5 cm) to 0.33 and 1.16 Mg C ha$^{-1}$ yr$^{-1}$ for the whole 1 m soil profile at DTC and UZF,

respectively. The depth and treatment interaction had no significant (p > 0.05) effects at both sites."

L384 so also provide likewise figures for UFZ

We agree with the comment and changes were addressed to include the average SOC accumulation rates for the top soil as well as the whole soil profile for UZF (line 430-432).

L391 'had' instead of 'has'; overall the sentences in this section 387-398 read strangely 'treatment x had net loss or accumulation of SOC' rephrase please in a more active form.

Comments were addressed as suggested and the paragraph now reads as follows (line 434-446):

"On the other hand, the different treatments in this study had significant (p < 0.05) effects in SOC accumulation / loss rates in the top 20 cm soil layer at both sites (Table 2). At DTC, NT had significant (p < 0.05) net loss of SOC in the 0-20 cm layer, ranging between -0.09 and -0.02 Mg C ha$^{-1}$ yr$^{-1}$, whereas NT treatments (NTM, NTR, NTMR) had SOC accumulation rates ranging from 0.17 to 0.38 Mg C ha$^{-1}$ yr$^{-1}$. However, maize stover mulching (NTM) had significantly (p < 0.05) higher SOC accumulation rates than CTR (2.9 – 4.2 times) and NT (5.2 – 13.5 times) in the top 15 cm and 20 cm layers, respectively. The different combinations of mulching and rotation under NT (NTM, NTR and NTMR) had no significant (p > 0.05) differences in SOC accumulation rates. Similarly, rotation treatments (CTR, NTR, NTMR) showed no significant (p > 0.05) differences in SOC accumulation rates. Thus, the full CA treatment had similar SOC accumulations rates to treatments with at least two combinations of CA principles (NTM and NTR) and to CTR."

Use the same manner to denote significance level in Tables 1 & 2

We agree with the comment and it was addressed accordingly in Table 2.

L418 rewrite this sentence

We agree with the comment and the sentence (lines 473-474) was rephrased to read "There were significant (p < 0.001) differences in cumulative OC inputs between treatments (Figure 4).".

L430 '0.05' I assume?

The observation is correct, it was a typo- error. However, we have since omitted all $CO_2$ emission data and the section was deleted.

L465 instead of > 50%, write maybe 'over half'

The suggestion is acceptable. We have addressed accordingly to read as follows (lines 545-546):

"This means that, over half of SOC stocks for this study were in the sub-soil (30-100 cm) which reflects on the importance of sub-soil SOC stocks."

L469 'treatment effects' what sort of treatments?

The comment is noble. We have specified that we are referring to no-tillage (NT) plus mulch and/or rotation treatments. The statement now reads (lines 549-553):

"Significant effects of mulch and/or rotation under NT were restricted to the top 30 cm in our study as well as other studies in SSA (Dube et al., 2012; Powlson et al., 2016) and the world at large (Balesdent et al., 2018; Yost and Hartemink, 2020), which is most likely why the default soil depth for IPCC for SOC studies is 0-30 cm (IPCC, 2019)."

L472 long and difficult to understand sentence

We concur with the comment and we have split the sentence into two. We further condensed the second sentence after splitting as follows (line 554-560):

"However, this underestimates whole soil profile C storage (Harrison et al., 2011; Lorenz and Lal, 2005; Singh et al., 2018). Therefore, it is crucial to consider whole soil profile sampling when monitoring SOC storage in agricultural ecosystems to determine their C sequestration potential in the pursuit of climate change mitigation (Malepfane et al., 2022)".

L491 add 'and' before 'hence'

We addressed the suggested comment (line 622).

L497 'do not' instead of 'does not', 'add SOC' is also not very well phrased further on in the sentence

Comment was considered and the sentence was rephrased accordingly (lines 627-630) to read; "It has been reported that NT cropping systems enhance SOC accumulation through increasing C inputs in the top layers and reducing erosion through minimum soil disturbance (Six et al., 2000; Lal, 2015, 2018; Bai et al., 2019; Cai et al., 2022)."

L514 Be clear, supposedly you are referring to an N-fertilization effect onto maize, but that is not spelled out here. Do then also report on the found (in)differences in maize crop residue returns.

Thank you for the comment. We have clarified that there were no significant treatment effects due to maize-cowpea rotation, as expected from increased N cycling due to biological nitrogen fixation from the preceding cowpeas. Instead, at DTC, it was maize stover mulching that improved maize yields (Shumba et al., 2023b) and subsequently belowground biomass (lines 647-650).

L527-535 shorten and rewrite so that this reason for finding no difference in SOC stock when considering deeper soil profiles could potentially also be in part attributed to lack of replicates.

We took cognisance of the comment and we effected the suggested remark where we alluded that our study was limited to four replicates which might not have enough statistical power to detect significant differences between treatments (lines 584-593).

L545 best to refer once more to DTC here

We addressed the suggested remark (line 526).

**Referee 1 (RC1)**

The paper is very well written, the trial is comprehensive, and methods were explained in detail. The high level of detail in the data presented is a rare find, and valuable insight in the understanding around SOC dynamics in CA systems. I was unable to find any problems in the manuscript, as far as I can tell the trial layout, sampling and analysis are correct, and the manuscript is written in clear language, that convey the message effectively.

We thank the Reviewer for sparing some time to review our manuscript and we are grateful for the overall positive and encouraging comment!

**Referee 2 (RC2)**

An overall relevant contribution on use of conservation agriculture practices to improve maize crop productivity and raise carbon stocks (and in turn fertility) in Zimbabwe. The study relies on two well-established field trials and standard yet well executed assessments were used to reveal how tillage, mulching and crop rotation in isolation or jointly influence SOC in these understudied systems. While the approach is overall robust – aside from limited attention to crop C inputs – at times the interpretation remains rather superficial. I have listed a number of comments below:

L103 & 108 One aspect of the hypotheses is best elaborated a bit more: while it is clear that crop residue retention and adoption of no-tillage are likely to improve SOC stocks, this is less obvious so for crop rotation. Crop rotation as such rather seems like a means to combat pests, but why should there necessarily be any beneficial effect onto SOC? Indirectly perhaps, via enhanced crop growth?

We agree with the reviewer's comment and we have now better explained this in the introduction section (lines 117-127). Below is the summary of the hypotheses.

Crop rotation is indeed crucial to reduce pests and diseases. The hypothesis is that the productivity of the crops is enhanced due to this reduction is biotic pressure, and therefore carbon inputs to the soil might be increased too. The second point is that this rotation introduces a nitrogen fixing crop, cowpea. We expect the following maize crop to benefit from this additional soil nitrogen. The third hypothesis is that, crop diversification enhances soil biological processes via different root systems, and possibly enhanced microfauna diversity and/or abundance (e.g. mycorrhizae) that could improve aggregate stability and therefore physical protection of soil carbon. The last hypothesis is that high quality residues (from the legume crop) have been shown to be preferentially stabilized in the soil due to a higher carbon use efficiency of soil microbes (Cotrufo et al., 2013; Kopittke et al., 2018).

L300 Why carry out statistical comparisons for each individual date? This does not provide an overall view on the treatment effects and complicates drawing conclusions.

We have since decided to omit the $CO_2$ emission data from this manuscript as suggested by the Topic Editor. Nonetheless, it is our pleasure to explain why we carried out statistical comparisons for each date.

Cumulative $CO_2$ emissions per treatment and per hectare are computed using a linear interpolation between daily $CO_2$ efflux. Using sampling dates as a fixed effect in the linear mixed effect model does not prevent us from comparing emissions per treatment, it just gives an additional information on periods of $CO_2$ efflux. Using daily emissions in statistical models is a common practice and allows a comparison based on measurements and not on interpolation, which we think is more accurate (Kravchenko and Robertson, 2015; Barthel et al., 2022;Shumba et al., 2023).

In the UFZ site, Fig 3 reveals a rather conspicuous treatment effect of including a rotation 90cm depth (and to a lesser extent at 60cm as well). Perhaps relevant to discuss as well, even though these trends were apparently not (yet) significant. Maybe the introduction of peas particularly led to deeper root proliferation and derived C-inputs at depth… perhaps the authors could try

using contrasts in the ANOVA to pool the three 'rotation' treatments vs. the other treatments and demonstrate such effect?

We agree with the observations by the Reviewer. However, there is a block effect at UZF (blocks 3 and 4) on SOC concentration in deep soils as shown in Figure R1 below. We decided not to exclude the "outliers" since we thought it was a block effect rather than a treatment effect, all treatments are affected. As a result, there are no significant differences in SOC concentration between treatments in deep soils. If we exclude the "outliers" in deep soils the graph for SOC concentration is as shown in Figure R2. All raw data of this paper can be freely accessed on the CIRAD database and linked to this paper (https://doi.org/10.18167/DVN1/VPOCHN). We have included the link to the raw data in the materials and methods section (Section 2) and in the Data analysis sub-section (Section 2.6).

We have now also added these Figures R1 and R2 in supplementary materials (Figure S2 and S3, respectively) and discussed the implications in the discussion section. At this stage we have excluded the addition of error bars on Fig 3 as the graphs will be too congested. Moreover, we have improved the axis of Fig 3 as suggested by the Topic Editor.

[Figure]

**Figure R1:** SOC concentration for the 100 cm soil profile for each treatment and replicate (block) showing "outliers" mainly in the 3rd and 4th replicates (blocks) at UZF.

[Figure]

Figure R2: Mean SOC concentration for the 100 cm soil profile for each treatment excluding "outliers".

The discussion section requires improvement:

Thank you for the comment and we have tackled the respective comments below.

The second half of 4.1 does not make much of a connection with own observations (L476-483) – it is not clear what message the authors wanted to bring here.

We agree with observations by the Reviewer we have rephrased the section which is now Section 4.2 (lines 560-567) to read as follows:

"SOC mineralization is relatively low in the sub-soil due to lack of oxygen and physical protection of SOC (aggregate protected C) (Rumpel et al., 2012; Sanaullah et al., 2016; Shumba et al., 2020; Button et al., 2022). Therefore, in the pursuit to improve subsoil (> 30 cm) SOC stocks through root mortality and exudates, crop varieties with higher root-length densities (Chikowo et al., 2003) in the subsoil are recommended."

However, in the section referenced, we were discussing the implications of our study findings as well as suggesting alternative ways of increasing sub-soil SOC stocks. The main message we were putting across is the importance of sub-soils in SOC storage as well as stressing whole soil profile sampling in SOC monitoring studies.

Section 4.2 is lengthy - Perhaps this section could benefit from further subdivision into subparts that deal with the various treatment aspects (tillage, mulching, crop rotation).

We agree with the Reviewer's comments and we have subdivided Section 4.2 which is now Section 4.3 (lines 578-684) accordingly into three sub-sections which delt with the different treatment aspects in our study. Sub-section 4.3.1, 4.3.2 and 4.3.3 delt with mulching, tillage and maize-cowpea rotation aspects, respectively. Some editorials and references were also added to improve the section.

4.2 lacks a clear calculation of conversion efficiency of C-inputs into SOC. There have been ample studies on removal or incorporation of maize residues (in light of research into the value of root vs. shoot C inputs) – such studies have generally revealed that aboveground biomass C inputs are far less effective in sustaining SOC – and this body of literature should be used to complement the discussion here. Even though much of the C-inputs are just derived from yield data, it would still be relevant to see some estimates of the efficiency by which these C-inputs formed or sustained SOC.

Thank you for the suggestion to calculate conversion efficiency of C-inputs into SOC. We tried to calculate this efficiency in the 0-30 cm depth for some treatments only. It was indeed not possible to calculate it for treatments with lower SOC stocks than the reference, or for treatments having higher SOC stocks but with less C-inputs than the reference (for example NTR at DTC). Here is the result:

| Treatments | DTC | UZF |
|------------|-----|-----|
| NTM vs CT | 0.34 | 0.46 |
| NTM vs NT | 0.45 | 0.27 |
| NTMR vs CT | 0.75 | 0.17 |
| NTMR vs NT | 0.98 | |

Some of these rates are unrealistic, especially the rates of NTMR at DTC. This is due to the high uncertainty linked to the estimation of C inputs to the soil, but also to the low number of replicates (4) for SOC stocks. Another explanation could be that most additional SOC might be in the form of labile particulate organic matter < 2mm, i.e decomposing plant materials accumulating in the soil, but not really stabilized carbon. While the calculation of the conversion efficiency of C-inputs into SOC could have been of interest, we believe they are too uncertain to show them. We have therefore decided not to include them in the manuscript.

We also took cognisant of the comments raised by the Reviewer on aboveground- vs belowground biomass in sustaining SOC. We have discussed the essence of belowground biomass to SOC storage in Section 4.3 (sub-section 4.3.1) and giving relevant literature (Hirte et al., 2021, 2018; Jones et al., 2009; Villarino et al., 2021) in support of the discussion (lines 595-614).

The discussion lacks considerations on the potential mechanisms through which the inclusion of peas in the crop rotation might influence maize C inputs. With a legume introduced there

may well be more N-available for the maize crop – not trivial given the very low doses of mineral N-supplied. If so, we may expect this to hold an effect on the soil C balance not only through an overall stimulation of maize productivity (and that is currently not clearly explained in the current discussion) but possibly also by effects onto the maize rooting pattern. These aspects should at least be commented upon, especially in light of the rather conspicuous difference in SOC at 75 or 90cm depth when a rotation is included (at least such would seem so from Fig 3, but unfortunately no comparison is given of the 75-100cm SOC stocks).

In concurring with the Reviewer, we have included the nitrogen input through biological nitrogen fixation from cowpeas in stimulating productivity of the succeeding maize crop (Sub-section 4.3.3, lines 653-668). We also included another potential SOC stabilization mechanism in the form of addition of high-quality cowpeas residues (above-and belowground biomass) which have been shown to be preferentially stabilized in the soil due to a higher carbon use efficiency of soil microbes (Cotrufo et al., 2013; Kopittke et al., 2018).

In our case, we think the possible rotation effects in deep soil SOC stocks at UZF is an artefact due to a block effect as explained above with graphical illustrations (Figures R1 and R2) above. We have now also discussed that.

Towards the end of the discussion and in the conclusion clearly the specificity of the here established crop rotation needs to be stipulated. By no means could we simply extrapolate found results to other 'crop rotations' – with different crops and plant C-inputs.

We agree with the comment and we have improved the section where we discussed the issue of OC quality in terms of C:N ratio where there was alternate addition of low- (maize) and high- (cowpea) quality OC inputs in the cowpea-maize rotation treatment especially in sub-section 4.3.3. We further discussed the relevance of these rotation systems which had significant effects on SOC stocks under NT systems albeit under soils with higher clay content typical of UZF in our study (lines 661-668).

L497: what about the whole idea that reduction of tillage would promote physical protection of OM inside microaggregates – should be commented on here as well.

We agree with the comment and we have included the suggested idea in the discussion (sub-section 4.3.2, lines 630-632) with some references to support it. However, the part which was commented by the reviewer has been restructured to read as follows (lines 627-639);

"It has been reported that NT cropping systems enhance SOC accumulation through increasing C inputs in the top layers and reducing erosion through minimum soil disturbance (Six et al., 2000; Lal, 2015, 2018; Bai et al., 2019; Cai et al., 2022). Minimum soil disturbance through NT also physically protects SOC in microaggregates from exposure to oxidative losses (Shumba et al., 2020; Six et al., 2002; Dolan et al., 2006; Liang et al., 2020). However, NT without mulch is a nonentity compared to other combinations of CA principles for long-term sustainability in cropping systems (Nyamangara et al., 2013; Kodzwa et al., 2020; Mhlanga et al., 2021; Li et al., 2020; Bohoussou et al., 2022) and NT is only effective in increasing SOC stocks when it is associated with other CA principles, especially mulch. On the other hand, our study suggest that NT can achieve the same results of SOC storage as NTR and NTMR since they had similar SOC stocks at UZF. This can be explained by the low aboveground OC inputs in rotation treatments during the season when cowpeas were grown."

L510 SOC storage does not seem very site specific but rather rotation-specific + 'the differences in the response to SOC changes' makes little sense -> rephrase

We agree with the comment and we have deleted the sentence in pursuit of compressing the section (lines 640-642).

L518 'the net SOC loss in CTR was due to seasonal exposure to oxidative losses (SOC mineralization) through disruption of soil macroaggregates by tillage' this makes little sense since these numbers are actually obtained by comparison with the CT treatment in which there is an equally intensive disruption of soil structure.

In this case we were referring to the UZF site (now indicated in line 621), for the comparison of CTR vs NTR where there were significant differences in SOC accumulation / loss rates. At DTC, it was the mulching component that had significant effects on SOC accumulation / loss rates (Table 2). Below are the new phrases we used to elaborate our point (lines 658-668).

"However, the net SOC loss in CTR at UZF was due to seasonal exposure to oxidative losses (SOC mineralization) through disruption of soil macroaggregates by tillage as alluded by Bai et al., (2019); Cambardella and Elliott, (1993) and Lal, (2018). We underscore that maize-cowpea rotation under NT improved SOC accumulation in the top soil due to reduced soil disturbance and alternate OC inputs of high (cowpeas) and low quality (maize). High quality OC inputs have a positive priming effect (Chen et al., 2014) which have been shown to be preferentially stabilized in the soil due to a higher carbon use efficiency of soil microbes (Cotrufo et al., 2013; Kopittke et al., 2018). This explains significant improvement in SOC stocks under the combination of NT and alternate high- and low-quality OC inputs (maize-cowpea rotation) to the soil in medium to heavy textured soils at UZF and vice versa at DTC."

L547 what about other SOC stabilization mechanisms?

We have added SOC adsorption to clay particles as another possible SOC stabilization mechanism (lines 532-534).

4.4 as it is stands alone and really does not bring a clear message. It may be better to integrate this section into 4.2 (which then gets even lengthier and is therefore best split).

Thank you for the suggestion, we have taken account of your comments and the Topic Editor's and resolved to delete $CO_2$ emission data and literature in all sections.

The conclusion section is fine.

---

## Author Response (AR2)

The authors have appropriately modified their previously submitted draft manuscript as per the reviewers' and editor's comments.

We are very grateful to your comments and we have addressed them below. In black is the Topic Editor's comments and our responses to the comments are in blue. In brief, we agree with the suggested comments and we have addressed them accordingly. The line numbers being referred to in this document corresponds to the new numbering in the manuscript with tracked changes.

Just a few minor changes to be made:

In the revised version now several potential mechanisms are forwarded via which inclusion of a leguminous crop in the rotation could promote storage of SOC. These concern reduced pest damage and N-provision among others. Since these particular effects were not measured in this study it would be better to not use the term 'hypothesize' or 'hypothesis' in L116-L126 as that raises expectations and readers will be searching for a testing of these hypotheses. Also, the first sentence is overly long (L116) and complex. Do rephrase with conditional verbs.

We are grateful to the Topic Editor on the comment and we have edited the section as suggested (Lines 107 – 123) which now reads as follows:

"We hypothesized that the full combination of CA components would be associated with higher increases in SOC stocks than adoption of only one component. This increase in SOC stocks could mainly be due to increased C inputs to the soil, especially under minimum soil disturbance. However, C inputs due to crop rotation could be indirect through increased crop productivity due to reduction on biotic pressure (pests and diseases), and therefore C inputs to the soil might be increased too. Cereals in a cereal-legume rotations may benefit from added soil nitrogen through biological nitrogen fixation from the preceding legume crop enhancing their productivity. Crop diversification, on the other hand, can enhance soil biological processes by increasing the diversity and/or abundance of microfauna like mycorrhizae. This, in turn, improves aggregate stability and offers physical protection for SOC. Lastly, high quality residues (from the legume crop) have been shown to be preferentially stabilized in the soil due to a higher carbon use efficiency of soil microbes (Cotrufo et al., 2013; Kopittke et al., 2018)".

For the recompiled figures: do not use serif-type fonts (like times new roman) but rather a font like arial or calibri, as you in fact did implement in Fig. 3

We agree to the comment and we have rectified and used Arial font (Lines 287 – 288)

Perhaps moving 4.1 (previously 4.3) forward was not the best choice. By doing so, the discussion now abruptly starts with a general statement on soil texture, then followed by L523-526 stating that soil textural differences between both sites explain the found differences in SOC storage and efficiency of retainment of crop-C inputs. This order is not optimal. Do instead start off by explaining, based on the results, first and foremost that there was a contrast in SOC stock between both sites. Then compare crop residue inputs between the sites and apparent efficiency thereof to form or sustain SOC in topsoil. Then bring in soil texture as likely explanation. Otherwise do perhaps reconsider 4.1s position in the discussion.

We have taken cognisance of the comment and we have reverted to the previous arrangement as suggested by the Topic Editor (Lines 401 – 566).

L568-577, this newly added text will confuse readers and is not needed. Do remove from the manuscript; Your explanation in the response letter sufficed and there was no need to also elaborately comment in the text itself.

We agree to the comment and we have deleted the paragraph (Lines 446 – 455).